# Connecting omics signatures and revealing biological mechanisms with iLINCS

Marcin Pilarczyk [1,2,3,4,9], Mehdi Fazel-Najafabadi [1,2,3,4,9], Michal Kouril[2,3,4,5,9], Behrouz Shamsaei [1,2,3,4,9], Juozas Vasiliauskas[1,2,3,4], Wen Niu[1,2,3,4], Naim Mahi[1,2,3,4], Lixia Zhang [1,2,3,4], Nicholas A. Clark [1,2,3,4], Yan Ren[1,2,3,4], Shana White[1,2,3,4], Rashid Karim [1,6], Huan Xu[1,2,3,4], Jacek Biesiada[1], Mark F. Bennett [1,2,3,4], Sarah E. Davidson[1], John F. Reichard[1,2,3,4], Kurt Roberts[1], Vasileios Stathias[2,3,4,7], Amar Koleti[2,3,4,7], Dusica Vidovic[2,3,4,7], Daniel J. B. Clarke[2,3,4,8], Stephan C. Schürer [2,3,4,7], Avi Ma'ayan [2,3,4,8], Jarek Meller [1,2,3,4,5,6] & Mario Medvedovic [1,2,3,4] ✉

There are only a few platforms that integrate multiple omics data types, bioinformatics tools, and interfaces for integrative analyses and visualization that do not require programming skills. Here we present iLINCS (http://ilincs.org), an integrative web-based platform for analysis of omics data and signatures of cellular perturbations. The platform facilitates mining and re-analysis of the large collection of omics datasets (>34,000), pre-computed signatures (>200,000), and their connections, as well as the analysis of user-submitted omics signatures of diseases and cellular perturbations. iLINCS analysis workflows integrate vast omics data resources and a range of analytics and interactive visualization tools into a comprehensive platform for analysis of omics signatures. iLINCS user-friendly interfaces enable execution of sophisticated analyses of omics signatures, mechanism of action analysis, and signature-driven drug repositioning. We illustrate the utility of iLINCS with three use cases involving analysis of cancer proteogenomic signatures, COVID 19 transcriptomic signatures and mTOR signaling.

Transcriptomics and proteomics (omics) signatures in response to cellular perturbations consist of changes in gene or protein expression levels after the perturbation. An omics signature is a high-dimensional readout of cellular state change that provides information about the biological processes affected by the perturbation and perturbation-induced phenotypic changes of the cell. The signature on its own provides information, although not always directly discernable, about

the molecular mechanisms by which the perturbation causes observed changes. If we consider a disease to be a perturbation of the homeostatic biological system under normal physiology, then the omics signature of a disease are the differences in gene/protein expression levels between disease and non-diseased tissue samples.

The low cost and effectiveness of transcriptomics assays[1–4] have resulted in an abundance of transcriptomics datasets and signatures.

[1]Department of Environmental and Public Health Sciences, University of Cincinnati, Cincinnati, OH 45220, USA. [2]LINCS Data Coordination and Integration Center (DCIC), Cincinnati, USA. [3]LINCS Data Coordination and Integration Center (DCIC), New York, USA. [4]LINCS Data Coordination and Integration Center (DCIC), Miami, USA. [5]Division of Biomedical Informatics, Cincinnati Children's Hospital Medical Center, Cincinnati, OH 45229, USA. [6]Department of Electrical Engineering and Computer Science, University of Cincinnati, Cincinnati, OH 45220, USA. [7]Department of Molecular and Cellular Pharmacology, Miller School of Medicine and Center for Computational Science, University of Miami, Miami FL 33136, USA. [8]Department of Pharmacological Sciences, Mount Sinai Center for Bioinformatics, Icahn School of Medicine at Mount Sinai, New York, NY 10029, USA. [9]These authors contributed equally: Marcin Pilarczyk, Mehdi Fazel-Najafabadi, Michal Kouril, Behrouz Shamsaei. ✉e-mail: Medvedm@ucmail.uc.edu

Recent advances in the field of high-throughput proteomics made the generation of large numbers of proteomics signatures a reality[5,6]. Several recent efforts were directed at the systematic generation of omics signatures of cellular perturbations[7] and at generating libraries of signatures by re-analyzing public domain omics datasets[8,9]. The recently released library of integrated network-based cellular signatures (LINCS)[7] L1000 dataset generated transcriptomic signatures at an unprecedented scale[2]. The availability of resulting libraries of signatures opens exciting new avenues for learning about the mechanisms of diseases and the search for effective therapeutics[10].

The analysis and interpretation of omics signatures has been intensely researched. Numerous methods and tools have been developed for identifying changes in molecular phenotypes implicated by transcriptional signatures based on gene set enrichment, pathway, and network analyses approaches[11–13]. Directly matching transcriptional signatures of a disease with negatively correlated transcriptional signatures of chemical perturbations (CP) underlies the *Connectivity Map* (CMAP) approach to identifying potential drug candidates[10,14,15]. Similarly, correlating signatures of chemical perturbagens with genetic perturbations of specific genes has been used to identify putative targets of drugs and chemical perturbagens[2].

To fully exploit the information contained within omics signature libraries and within countless omics signatures generated frequently and constantly by investigators around the world, new user-friendly integrative tools, accessible to a large segment of biomedical research community, are needed to bring these data together. The integrative LINCS (iLINCS) portal brings together libraries of precomputed signatures, formatted datasets, connections between signatures, and integrates them with a bioinformatics analysis engine and streamlined user interfaces into a powerful system for omics signature analysis.

## Results

iLINCS (available at http://ilincs.org) is an integrative user-friendly web platform for the analysis of omics (transcriptomic and proteomic) datasets and signatures of cellular perturbations. The key components of iLINCS are: Interactive and interconnected analytical workflows for the creation and analysis of omics signatures; The large collection of datasets, precomputed signatures, and their connections; And user-friendly graphical interfaces for executing analytical tasks and workflows.

The central concept in iLINCS is the omics signature, which can be retrieved from the precomputed signature libraries within the iLINCS database, submitted by the user, or constructed using one of the iLINCS datasets (Fig. 1a). The signatures in iLINCS consist of the differential gene or protein expression levels and associated *P* values between perturbed and baseline samples for all, or any subset of measured genes/proteins. Signatures submitted by the user can also be in the form of a list of genes/proteins, or a list of up- and down-regulated genes/proteins. iLINCS backend database contains >34,000 processed omics datasets, >220,000 omics signatures and >10$^9$ statistically significant "connections" between signatures. Omics signatures include transcriptomic signatures of more than 15,000 chemicals and genetic perturbations of more than 4400 genes (Fig. 1a). Omics datasets available for analysis and signatures creation cover a

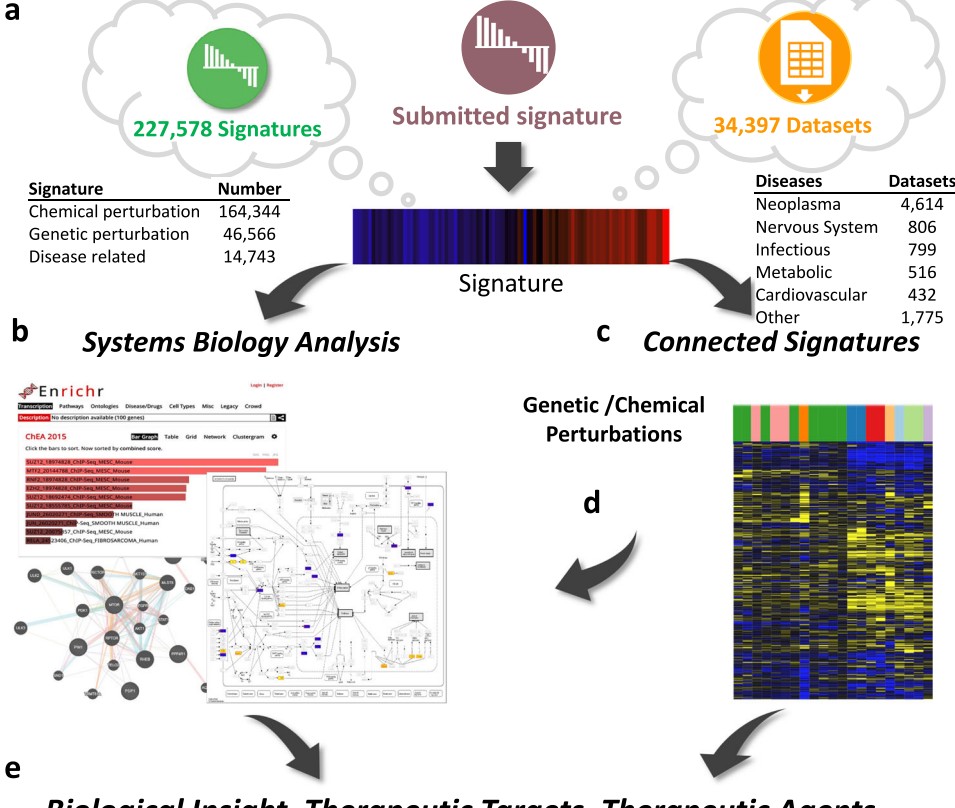

**Fig. 1 | Integrative omics signature analysis in iLINCS. a** A signature can be selected by querying the iLINCS database, submitted by the user, or constructed by analyzing an iLINCS omics dataset. Signatures in the database include chemical and genetic perturbation, and a wide range of disease-related signatures. The datasets cover a wide range of human diseases. **b** The signature can be analyzed using a range of systems biology methods (gene set enrichment, pathway and network analyses). **c** Signature "connectivity" analyses can be applied to identify cellular perturbations and biological states of similar signatures. **d** The analysis of connected signatures, as well as the identity of the perturbed genes and proteins leading to the connected signatures, can be used to elucidate mechanisms of action. **e** Ultimately, the results of the analyses lead to insights and hypotheses about potential therapeutic targets and therapeutic agents.

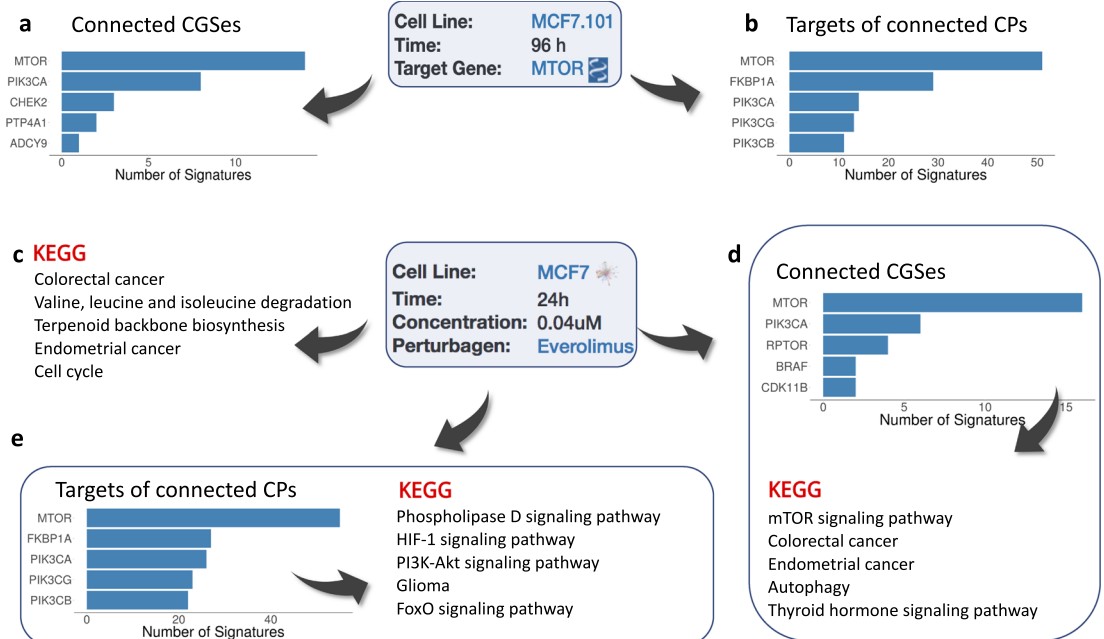

**Fig. 2 | Analysis of LINCS L1000 signatures of genetic and chemical perturbations. a** Most frequently perturbed genes among the Consensus Genes Signatures (CGS) connected to the mTOR knockdown CGS. **b** Most frequent inhibition targets of chemical perturbagens with signatures connected to the mTOR CGS signature. **c** Most enriched biological pathways for the everolimus signature. **d** Most frequently perturbed genes among CGSes connected with everolimus signature, and pathways most enriched by the perturbed genes. **e** Most frequent inhibition targets of chemical perturbagens with signatures connected to the everolimus signature and the pathways most enriched by the genes of the targeted proteins.

wide range of diseases and include transcriptomic (RNA-seq and microarray) and proteomic (Reverse Phase Protein Arrays[16] and LINCS-targeted mass spectrometry proteomics[5]) datasets. Datasets collections include close to complete collection of GEO RNA-seq datasets and various other dataset collections, such as The Cancer Genome Atlas (TCGA), GEO GDS microarray datasets[17], etc. A detailed description of iLINCS omics signatures and datasets is provided in "Methods". Analysis of 8942 iLINCS datasets from GEO, annotated by MeSH terms[18], shows a wide range of disease coverage (Fig. 1a).

iLINCS analytical workflows facilitate systems biology interpretation of the signature (Fig. 1b) and the connectivity analysis of the signature with all iLINCS precomputed signatures (Fig. 1c). Connected signatures can further be analyzed in terms of the patterns of gene/protein expression level changes that underlie the connectivity with the query signature, or through the analysis of gene/protein targets of connected perturbagens (Fig. 1d). Ultimately, the multi-layered systems biology analyses, and the connectivity analyses lead to biological insights, and identification of therapeutic targets and putative therapeutic agents (Fig. 1e).

Interactive analytical workflows in iLINCS facilitate signature construction through differential expression analysis as well as clustering, dimensionality reduction, functional enrichment, signature connectivity analysis, pathway and network analysis, and integrative interactive visualization. Visualizations include interactive scatter plots, volcano and GSEA plots, heatmaps, and pathway and network node and stick diagram (Supplemental Fig. 1). Users can download raw data and signatures, analysis results, and publication-ready graphics. iLINCS internal analysis and visualization engine uses R[19] and open-source visualization tools. iLINCS also facilitates seamless integration with a wide range of task-specific online bioinformatics and systems biology tools and resources including Enrichr[20], DAVID[21], ToppGene[22], Reactome[23], KEGG[24], GeneMania[25], X2K Web[26], L1000FWD[27], STITCH[28], Clustergrammer[29], piNET[30], LINCS Data Portal[31], ScrubChem[32], PubChem[33] and GEO[34]. Programmatic access to iLINCS data, workflows and visualizations are facilitated by the calls to iLINCS API which is

documented with the OpenAPI community standard. Examples of utilizing the iLINCS API within data analysis scripts are provided on GitHub (https://github.com/uc-bd2k/ilincsAPI). The iLINCS software architecture is described in Supplemental Fig. 2.

**Use cases**
iLINCS workflows facilitate a wide range of possible use cases. Querying iLINCS with user-submitted external signatures enables identification of connected perturbations signatures, and answering in-depth questions about expression patterns of individual genes or gene lists of interest in specific datasets, or across classes of cellular perturbations. Querying iLINCS with individual genes or proteins can identify sets of perturbations that significantly affect their expression. Such analysis leads to a set of chemicals, or genetic perturbations, that can be applied to modulate the expression and activity of the corresponding proteins. Queries with lists of genes representing a hallmark of a specific biological state or process[35] can identify a set of perturbations that may accordingly modify cellular phenotype. iLINCS implements complete systematic polypharmacology and drug repurposing[36,37] workflows, and has been listed as a Bioinformatics resource for cancer immunotherapy studies[38] and multi-omics computational oncology[39]. Most recently, iLINCS has been used in the drug repurposing workflow that combines searching for drug repurposing candidates via CMAP analysis with the validation using analysis of Electronic Health Records[40]. Finally, iLINCS removes technical barriers for re-using any of more than 34,000 preprocessed omics datasets enabling users to construct and analyze new omics signatures without any data generation and with only a few mouse clicks.

Here, we illustrate the use of iLINCS in detecting and modulating aberrant mTOR pathway signaling, analysis of proteogenomic signatures in breast cancer and in search for COVID-19 therapeutics. It is important to emphasize that all analyses were performed by navigating iLINCS GUI within a web browser, and each use case can be completed in less than five minutes. Step-by-step instructions are provided in the Supplemental Materials (Supplemental Workflows 1, 2, and 3). In

addition, links to instructional videos that demonstrate how to perform these analyses are provided on the landing page of iLINCS at ilincs.org. The same analyses can also be performed programmatically using the iLINCS API. R notebooks demonstrating this can be found on the GitHub (https://github.com/uc-bd2k/ilincsAPI).

## Use case 1: detecting and modulating aberrant mTOR pathway signaling

Aberrant mTOR signaling underlies a wide range of human diseases[41]. It is associated with age-related diseases such as Alzheimer's disease[42] and the aging process itself[41]. mTOR inhibitors are currently the only pharmacological treatment shown to extend lifespan in model organisms[43], and numerous efforts in designing drugs that modulate the activity of mTOR signaling are under way[41]. We use mTOR signaling as the prototypical example to demonstrate iLINCS utility in identifying chemical perturbagens capable of modulating a known signaling pathway driving the disease process, in establishing MOA of a chemical perturbagen, and in detecting aberrant signaling in the diseased tissue. Detecting changes in mTOR signaling activity in transcriptomic data is complicated by the fact that it is not reflected in changes in expression of mTOR pathway genes, and standard pathway analysis methods are not effective[44]. We show that CMAP analysis approach, facilitated by iLINCS, is essential for the success of these analyses. Step-by-step instructions for performing this analysis in iLINCS are provided in Supplemental Workflow SW1.

Identifying chemicals that can modulate the activity of a specific pathway or a protein in a specific biological context is often the first step in translating insights about disease mechanisms into therapies that can reverse disease processes. Here we demonstrate the use of iLINCS in identifying chemicals that can inhibit the mTOR activity. We use the Consensus Genes Signatures (CGSes) of CRISPR mTOR genetic loss of function perturbation in MCF-7 cell line as the query signature. The CMAP analysis identifies 258 LINCS CGSes and 831 CP Signatures with statistically significant correlation with the query signature. Top 100 most connected CGSes are dominated by the signatures of genetic perturbations of mTOR and PIK3CA genes (Fig. 2a), whereas all top 5 most frequent inhibition targets of CPs among top 100 most connected CP signatures are mTOR and PIK3 proteins (Fig. 2b). Results clearly indicate that the query mTOR CGS is highly specific and sensitive to perturbation of the mTOR pathway and effectively identifies chemical perturbagens capable of inhibiting mTOR signaling. The full list of connected signatures is shown in Supplemental Data SD1. The connected CP signatures also include several chemical perturbagens with highly connected signatures that have not been known to target mTOR signaling providing new candidate inhibitors.

Identifying proteins and pathways directly targeted by a bioactive chemical using its transcriptional signature is a difficult problem. Transcriptional signatures of a chemical perturbation often carry only an echo of such effects since the proteins directly targeted by a chemical and the interacting signaling proteins are not transcriptionally changed. iLINCS offers a solution for this problem by connecting the CP signatures to LINCS CGSes and facilitating a follow-up systems biology analysis of genes whose CGSes are highly correlated with the CP signature. This is illustrated by the analysis of the perturbation signature of the mTOR inhibitor drug everolimus (Fig 2c–e). Traditional pathway enrichment analysis of this CP signature via iLINCS connection to Enrichr (Fig. 2c) fails to identify the mTOR pathway as being affected. In the next step, we first connect the CP signature to LINCS CGSes and then perform pathway enrichment analysis of genes with correlated CGSes. This analysis correctly identifies mTOR signaling pathway as the top affected pathway (Fig. 2d). Similarly, connectivity analysis with other CP signatures followed by the enrichment analysis of protein targets of top 100 most connected CPs again identifies the Pi3k-Akt signaling pathway as one of the most enriched (Fig. 2e). In conclusion, both pathway analysis of differentially

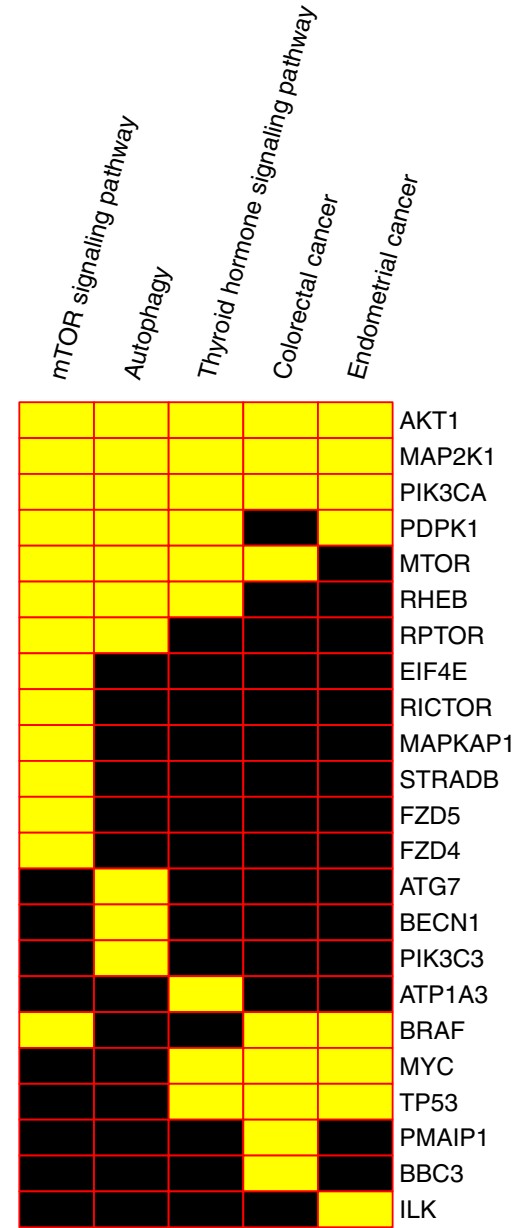

**Fig. 3 | Perturbation gene targets in enriched pathways.** Yellow squares indicate the membership of the target gene (rows) in the corresponding pathway (columns).

expressed genes in the everolimus signature and pathway analysis of connected genetic and chemical perturbagens provide us with important information about effects of everolimus. However, only the analyses of connected perturbagens correctly pinpoints the direct mechanism of action of the everolimus, which is the inhibition of mTOR signaling.

The connectivity-based pathway analysis shares methodological shortcomings with the standard enrichment/pathway analyses of lists of differentially expressed genes, such as, for example, overlapping pathways. While the MTOR pathway shows the strongest association with everolimus, other pathways were also significantly enriched. A closer examination of the results indicates that this is due to the core mTOR signaling cascade being included as a component of other pathways and many of the genes that drive the associations with other four most enriched pathways are common with the mTOR pathway (Fig. 3).

One caveat in the results presented above is that the LINCS signatures based on L1000 platform provide a reduced representation of

the global transcriptome consisting of expression levels of about 1000 "landmark" genes[2]. The landmark genes are selected in such a way that they jointly capture patterns of expression of majority genes in the genome and the computational predictions of expression of additional 12,000 genes are also made. The relatively low number of measured genes could sometimes adversely affect the gene expression enrichment analysis of poorly represented pathways. To establish that this is not the case for mTOR signaling, we repeated the MOA analysis using the whole genome transcriptional signature of the mTOR inhibitor sirolimus from the original CMAP dataset[15], which is also included in the iLINCS signature collection. Results of these analyses closely resemble the results with L1000 everolimus signature with connectivity analysis clearly pinpointing mTOR pathway and enrichment analysis of differentially expressed genes failing to do so (Supplemental Results 4).

To verify that mTOR signaling modulation is also detectable in complex tissues we used iLINCS to re-analyze the effect of rapamycin in aged rat livers[45] (GEO dataset GSE108978). The rapamycin signature was constructed by comparing expression profiles of livers in eight rapamycin-treated rats to the nine vehicle controls at 24 months of age (Fig. 4, heatmap). The signature correlated strongly with CP signatures of chemicals targeting mTOR pathway genes (Fig. 4, bar plot).

## Use case 2: proteo-genomics analysis of cancer driver events in breast cancer

Contrasting transcriptional and proteomic profiles of different molecular cancer subtypes has long been a hallmark of cancer omics data analysis when seeking targets for intervention[46]. Constructing signatures by comparing cancer with normal tissue controls usually results in a vast array of differences characteristic of any cancer (proliferation, invasion, etc.)[47], and are not specific to the driver mechanisms of the cancer samples at hand. On the other hand, comparisons of different cancer subtypes, as illustrated here, is effective in eliciting key driver mechanisms by factoring out generic molecular properties

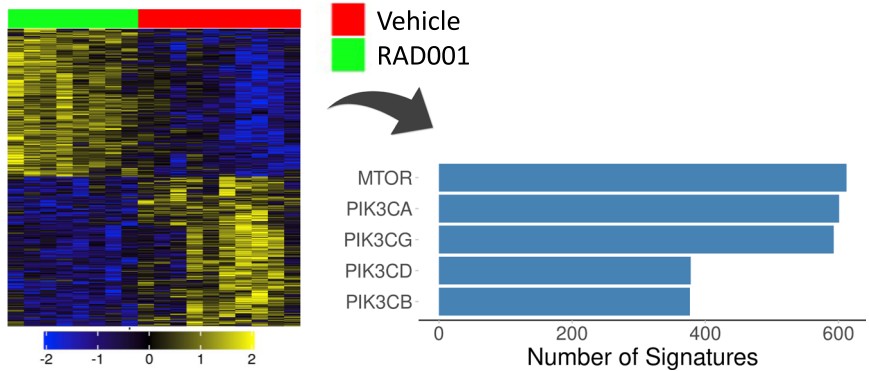

**Fig. 4 | CMAP analysis of rapamycin (RAD001) signature in rat livers.** The heatmap shows the centered expression levels of differentially expressed genes and the bar plot shows the numbers of connected chemical perturbation signatures for top five targets.

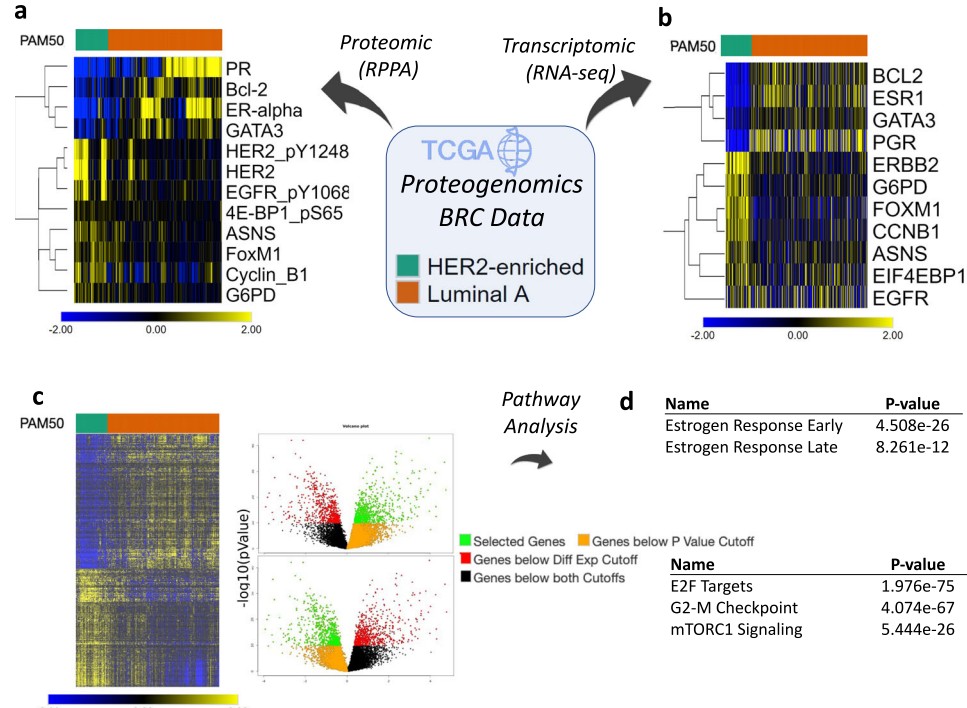

**Fig. 5 | Proteo-genomics analysis of cancer driver events in breast cancer. a** Most differentially expressed proteins in the proteomics signatures constructed by comparing RPPA profiles of Her2E and Luminal-A BRC samples. **b** Gene expression profile of the genes corresponding to proteins in (**a**) based on RNA-seq data. **c** The transcriptional signature consisting of all highly differentially expressed genes (unadjusted, two-tailed $P$ value<$10^{-10}$). **d** Enrichment analysis of genes upregulated in Luminal A, and upregulated in Her2E tumors via Enrich (unadjusted Fisher Exact Test $P$ values).

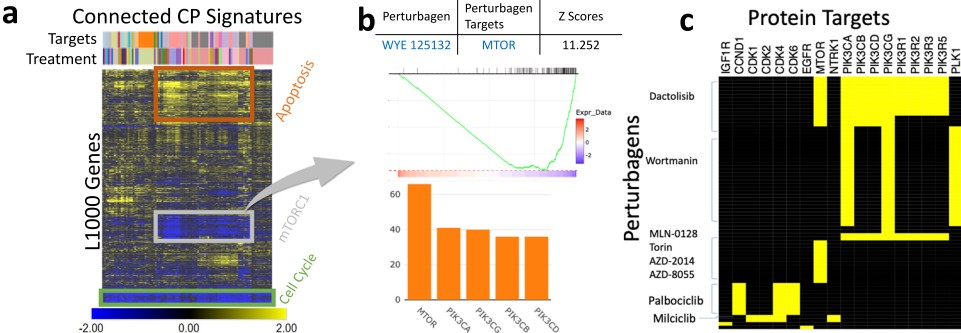

**Fig. 6 | Connectivity map analysis of Luminal A vs Her2E signatures. b** Top 100 connected CP signatures. **b** Signatures enriched for genes in the gray box. The GSEA plot for the most significantly enriched signature and the summary of targets for top 100 most enriched signatures. **c** Chemical perturbagens and their targets for CP signatures in (**a**).

of a cancer[48]. Here, we demonstrate the use of matched preprocessed proteomic (RPPA) and transcriptomic (RNA-seq) breast cancer datasets to identify driver events which can serve as targets for pharmacological intervention in two different breast cancer subtypes. The analysis of proteomic data can directly identify affected signaling pathways by assessing differences in the abundance of activated (e.g., phosphorylated) signaling proteins. By contrasting proteomics and transcriptomic signatures of the same biological samples, we can distinguish between transcriptionally and post-translationally regulated proteins, and transcriptional signatures facilitate pathway enrichment and CMAP analysis. Step-by-step instructions for performing this analysis in iLINCS are provided in Supplemental Workflow SW2.

We analyzed TCGA breast cancer RNA-seq and RPPA data using the iLINCS "Datasets" workflow to construct the differential gene and protein expression signatures contrasting 174 Luminal-A and 50 Her2 enriched (Her2E) breast tumors. The results of the iLINCS analysis track closely the original analysis performed by the TCGA consortium[48]. The protein expression signature immediately implicated known driver events, which are also the canonical therapeutic targets for the two subtypes: The abnormal activity of the estrogen receptor in Luminal-A tumors, and the increased expression and activity of the Her2 protein in Her2E tumors (Fig. 5a). To further validate the strategy of directly comparing two subtypes of tumors, we compared our results with the analysis results when different subtypes are compared to normal breast tissue controls (Supplemental Results 5). The result of these analyses are much more equivocal, with Her2 and ERalpha proteins now superseded in significance by several more generic cancer-related proteome alterations, common to both subtypes (Supplemental Results 5).

The corresponding Luminal A vs Her2E RNA-seq signature, constructing by differential gene expression analysis between 201 Luminal-A and 47 Her2E samples, showed similar patterns of expression of key genes (Fig. 5b). All genes were differentially expressed (Bonferroni adjusted *P* value < 0.01) except for EGFR, indicating that the difference in expression levels of the EGFR protein with the phosphorylated Y1068 tyrosine residue (EGFR_pY1068) may be a consequence of post-translation modifications instead of increased transcription rates of the EGFR gene.

Following down the iLINCS workflow, the pathway analysis of 734 most significantly upregulated genes in Luminal-A tumors (*P* value < 1e-10) (Fig. 5c) identified the Hallmark gene sets[35] indicative of Estrogen Response to be the most significantly enriched (Fig. 5d) (See Supplemental Data SD2 for all results). Conversely, the enrichment analysis of 665 genes upregulated in Her2E tumors identified the Hallmark gene sets of proliferation (E2F Targets, G2-M Checkpoint) and the markers of increased mTOR signaling (mTORC1 signaling). This reflects a known increased proliferation of Her2E tumors in comparison to Luminal-A tumors[49]. The increase in mTOR signaling is

consistent with the increased levels of the phosphorylated 4E-BP protein, a common marker of mTOR signaling[50].

The CMAP analysis of the RNA-seq signature with LINCS CP signatures (Fig. 6) shows that treating several different cancer cell lines with inhibitors of PI3K, mTOR, CDK, and inhibitors of some other more generic proliferation targets (e.g., TOP21, AURKA) (see Supplemental Data SD3 for complete results) produces signatures that are positively correlated with RNA-seq Luminal A vs Her2E signature, suggesting that such treatments may counteract the Her2E tumor driving events.

The detailed analysis of 100 most connected CP signatures showed that all signatures reflected proliferation inhibition as indicated by the enrichment of the genes in the KEGG Cell cycle pathway among the genes downregulated across all signatures (Fig. 6a). However, the analysis also showed that a subset of the signatures selectively inhibited expression of the mTORC1 signaling Hallmark gene set, and the same set of signatures exhibited increased upregulation of Apoptosis gene sets in comparison to the rest of the signatures. This indicates that the increased proliferation of in Her2E tumors may be partly driven by the upregulation in mTOR signaling.

We also used iLINCS to identify de novo all signatures enriched for the mTOR-associated genes from Fig. 6a. The most enriched signatures (top 100) were completely dominated by signatures of mTOR inhibitors (Fig. 6b). The most highly enriched signature was generated by WYE-125132, a highly specific and potent mTOR inhibitor[51]. Using the iLINCS signature group analysis workflow we also summarized the drug-target relationships for the top 100 signatures (Fig. 6c) which recapitulate the dominance of mTOR inhibitors along with proliferation inhibitors targeting CDK proteins (Palbociclib and Milciclib).

## Use case 3: drug repurposing for COVID-19

The ongoing COVID-19 pandemic has underscored the importance of rapid drug discovery and repurposing to treat and prevent emerging novel pathogens, such as SARS-CoV-2. As part of the community-wide efforts to identify novel targets and treatment options, the transcriptional landscape of SARS-CoV-2 infections has been characterized extensively, including the identification of transcriptional signatures from patients as well as model systems[52,53]. CMAP approach has been extensively used to explore that space of potential therapeutic agents with the search of Google Scholar website listing 662 studies for the covid AND "connectivity map" search. In iLINCS, 105 COVID-19-related datasets are organized into a COVID-19 collection, facilitating signature connectivity-based drug discovery and repurposing in this context.

We used iLINCS to construct a SARS-CoV-2 infection signature by re-analyzing the dataset profiling the response of various in vitro models to SARS-CoV-2 infection[52] (GEO dataset GSE147507). The use of multiple models, which respond differently to the virus infection, would make the signature created by direct comparisons of all

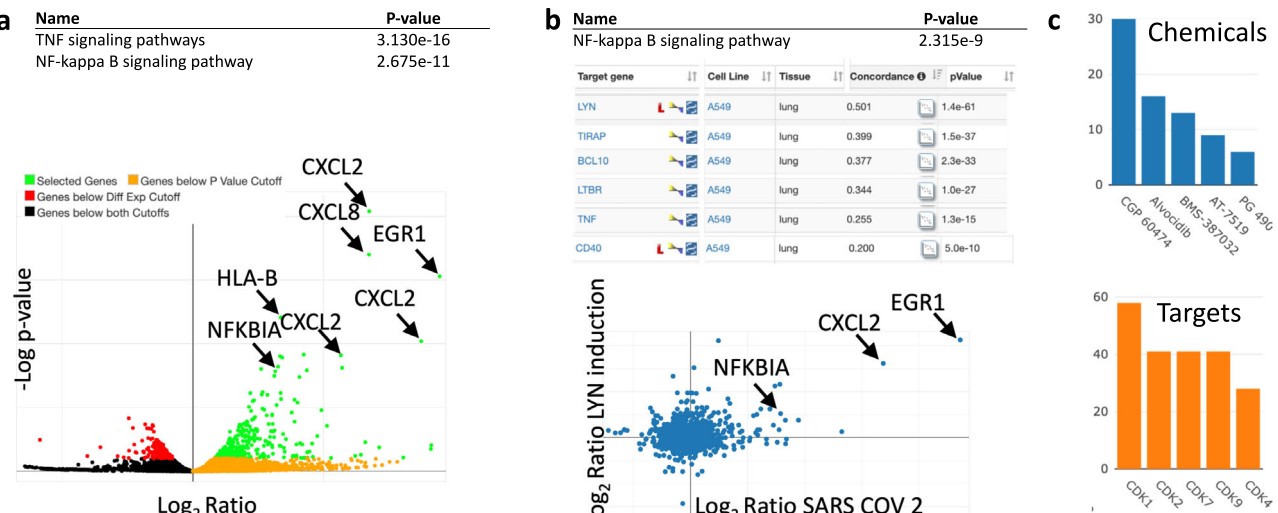

**Fig. 7 | SARS-Cov-2 infection of A549 cells expressing ACE2. a** Upregulated genes (unadjusted, two-tailed *P* value < $10^{-10}$) in top two enriched KEGG pathways (unadjusted Fisher exact test *P* values shown in the table). **b** Top KEGG pathway in the enrichment analysis (unadjusted Fisher exact test *P* values shown in the table) of signatures of gene overexpression mimicking infection in the A549 cell line. The list of six most positively correlated overexpression signatures (unadjusted, two-tailed weighted correlation *P* values are shown in the table) and the scatter plot of the LYN overexpression signature against the SARS Cov-2 infection signature. **c** Chemicals reversing the infection signatures and their protein targets.

"Infected" vs all "Mock infected" samples too noisy. The main mechanism implemented in iLINCS for dealing with various confounding factors is filtering samples by levels of possible confounding factors, which is the approach that is most used in the omics data analysis. In this case, we filtered samples to construct a signature by differential gene expression analysis of infected vs mock-infected A549 cell line, which was genetically modified to express *ACE2* gene to facilitate viral entry into the cell. This left us with the comparison of three "Infected" and three "Mock infected" samples. Filtering of samples and the analysis using iLINCS GUI is demonstrated in the Supplemental workflow SW3.

The resulting signature comprises many upregulated chemokines and other immune system-related genes, including the *EGR1* transcription factor that regulates inflammation and immune system response[54,55], and the pathway analysis implicates TNF signaling and NK-kappa B signaling as the two pathways most enriched for upregulated genes (Fig. 7a). CMAP analysis against the LINCS gene overexpression signatures in A549 cell line identified the signature of LYN tyrosine kinase as the most positively correlated with the SARS-CoV-2 infection signature (Fig. 7b). LYN is a member of SRC/FYN family of tyrosine kinases has been shown to be required for effective MERS-CoV replication[56]. The enrichment of genes with positively correlated overexpression signatures in A549 cell line, identified NF-kappa B signaling pathway as the most enriched (Fig. 7b), confirming mechanistically the role of NF-kappa B signaling in inducing the infection signature. Finally, CMAP analysis identified CDK inhibitors and the drug Avlodicip as the potential therapeutic strategies based on their ability to reverse the infection signature (Fig. 7c). Alvocidib is a CDK9 inhibitor with a broad antiviral activity and it has been suggested as a potential candidate for COVID-19 drug repurposing[57].

These results agree with the previous study utilizing iLINCS to prioritize candidate FDA-approved or investigative drugs for COVID-19 treatment[58]. Of the top 20 candidates identified in that study as reversing SARS-CoV-2 transcriptome signatures, 8 were already under trial for the treatment of COVID-19, while the remaining 12 had antiviral properties and 6 had antiviral efficacy against coronaviruses specifically. Our analysis illustrates the ease with which iLINCS can be used to quickly provide credible drug candidates for an emerging disease.

## Discussion

iLINCS is a unique integrated platform for the analysis of omics signatures. Several canonical use cases described here only scratch the surface of the wide range of possible analyses facilitated by the interconnected analytical workflows and the large collections of omics datasets, signatures, and their connections. All presented use cases were executed using only a mouse to navigate iLINCS GUI. Each use case can be completed in less than 5 min, as illustrated in the online help and video tutorials. The published studies to date used iLINCS in many different ways, and to study a wide range of diseases (Supplemental Results 3).

In addition to facilitating standard analyses, iLINCS implements innovative workflows for biological interpretation of omics signatures via CMAP analysis. In Use case 1, we show how CMAP analysis coupled with pathway and gene set enrichment analysis can implicate mechanism of action of a chemical perturbagen when standard enrichment analysis applied to the differentially expressed genes fails to recover targeted signaling pathways. In a similar vein, iLINCS has been successfully used to identify putative therapeutic agents by connecting changes in proteomics profiles in neurons from patients with schizophrenia; first with the LINCS CGSes of the corresponding genes, and then with LINCS CP signatures[59]. These analyses led to the identification of PPAR agonists as promising therapeutic agents capable of reversing bioenergetic signature of schizophrenia, which were subsequently shown to modulate behavioral phenotypes in rat model of schizophrenia[60].

The iLINCS platform was built with the flexibility to incorporate future extensions in mind. The combination of optimized database representation and R analysis engine provide endless opportunities to implement additional analysis workflows. At the same time, collections of omics datasets and signatures can be extended by simply adding data to backend databases. One of the important directions for improving iLINCS functionality will be the development of workflows for fully integrated analysis of multiple datasets and different omics data types. In terms of integrative analysis of matched transcriptomic and proteomic data, iLINCS facilitates the integration where results of one omics dataset informs the set of genes/proteins analyzed in the other dataset (Use case 2). However, the direct integrative analysis of both data types may result in more informative signatures[61]. At

the same time, addition of more proteomics datasets, such as the Clinical Proteomic Tumor Analysis Consortium (CPTAC) collection[62], will extend the scope of such integrative analyses.

Many complex diseases, including cancer, consist of multiple, molecularly distinct, subtypes[63–65]. Accounting for these differences is essential for constructing effective disease signatures for CMAP analysis. In Use case 2, we demonstrate how to use iLINCS in contrasting molecular subtypes when the information about the subtypes is included in sample metadata. An iLINCS extension that allows for de novo creation of molecular subtypes using cluster analysis, as is the common practice in analysis of cancer samples[63], is currently under development. Another future extension under development is the workflow for constructing disease signatures using single cell datasets[66]. iLINCS contains a number of single cell RNA-seq (scRNA-seq) datasets, but their analysis is currently handled in the same way as the bulk RNA-seq data. A specialized workflow for extracting disease signatures from scRNA-seq data will lead to more precise signatures and more powerful CMAP analysis[66].

With many signatures used in CMAP analysis, and a large number of genes perturbed by either genetic or chemical perturbations, one has to carefully scrutinize results of CMAP-based pathway analyses to avoid false positive results and identify most relevant affected pathways. Limitations of standard gene enrichment pathway analysis related to overlapping pathways are important to keep in mind in the signature-similarity-based pathway analysis, as discussed in Use case 1. In addition, the hierarchical nature of gene expression regulation may lead to similar transcriptional signatures being generated by perturbing genes at different levels of the regulatory programs (e.g., signaling proteins vs transcriptional factors). Perturbations of distinct signaling pathways leads to modulation of the proliferation rates in cancer cell lines, and it is expected that resulting transcriptional signatures share some similarities related to up- and down-regulation of proliferation drivers and markers. At the same time, signatures corresponding to perturbation of proteins regulating the same sets of biological processes are likely to exhibit a higher level of similarity. The analysis of the top 100 chemical perturbagen signatures negatively correlated with Her2E breast cancer signature in Use case 2 reveals that they all contain the "proliferation" component. However, a subset of the most highly correlated signatures is more specifically associated with mTOR inhibition, indicating that the proliferation is affected in part by modulating mTOR signaling. The association with perturbations that modulate cellular proliferation, while real, could also be considered spurious as it is relatively non-specific. The association with mTOR signaling is more specific and provides a higher-level mechanistic explanation for differences in proliferation rates. iLINCS provides mechanisms for scrutinizing expression profiles of genes in signatures identified in CMAP analysis that is required for assessing these fine points (Use case 2), and they are essential for interpreting the results of a CMAP analysis.

Several online tools have been developed for the analysis and mining LINCS L1000 signature libraries. They facilitate online queries of L1000 signatures[67–69] and the construction of scripted pipelines for in-depth analysis of transcriptomics data and signatures[70]. The LINCS Transcriptomic Center at the Broad Institute developed the *clue.io* query tool deployed by the Broad Connectivity Map team which facilitates connectivity analysis of user-submitted signatures[2]. iLINCS replicates the connectivity analysis functionality, and indeed, the equivalent queries of the two systems may return qualitatively similar results (see Supplemental Results 1 for a use case comparison). However, the scope of iLINCS is much broader. It provides connectivity analysis with signatures beyond Connectivity Map datasets and provides many primary omics datasets for users to construct their own signatures. Furthermore, analytical workflows in iLINCS facilitate deep

systems biology analysis and knowledge discovery of both, omics signatures and the genes and protein targets identified through connectivity analysis. Comparison to several other web resources that partially cover different aspects of *iLINCS* functionality are summarized in Supplemental Results 2.

iLINCS removes technical roadblocks for users without a programming background to re-use of publicly available omics datasets and signatures. The user interfaces are streamlined and strive to be self-explanatory to most scientists with conceptual understanding of omics data analysis. Recent efforts in terms of standardizing[71] and indexing[72] are improving findability and re-usability of public domain omics data. iLINCS is taking the next logical step in integrating public domain data and signatures with a user-friendly analysis toolbox. Furthermore, all analyses steps behind the iLINCS GUI are driven by API which can be used within computational pipelines based on scripting languages[73], such as R, Python and JavaScript, and to power the functionality of other web analysis tools[30,69]. This makes iLINCS a natural tool for analysis and interpretation of omics signatures for scientists preferring point-and-click GUIs as well as data scientists using scripted analytical pipelines.

## Methods

### Statistics
All differential expression and signature creation analyses are performed on measurements obtained from distinct samples. All *P* values are calculated using two-sided hypothesis tests. The specific tests depend on the data type and are described in detail in the rest of the methods and the Supplemental Methods document. The accuracy of the iLINCS datasets, signatures and analysis procedures were ascertained as described in the Supplemental Quality Control document. The versions of all R packages utilized by iLINCS are provided in the Supplemental Data SD4.

### Perturbation signatures
All precomputed perturbation signatures in iLINCS, as well as signatures created using an iLINCS dataset, consist of two vectors: the vector of log-scale differential expressions between the perturbed samples and baseline samples $\mathbf{d} = (d_1,...,d_N)$, and the vector of associated *P* values $\mathbf{p} = (p_1,...,p_N)$, where *N* is the number of genes or proteins in the signature. Signatures submitted by the user can also consist of only log-scale differential expressions without *P* values, lists of up- and downregulated genes, and a single list of genes.

### Signature connectivity analysis
Depending on the exact type of the query signature, the connectivity analysis with libraries of precomputed iLINCS signatures are computed using different connectivity metrics. The choice of the similarity metric to be used in different contexts was driven by benchmarking six different methods (Supplementary Result 2).

If the query signature is selected from iLINCS libraries of precomputed signatures, the connectivity with all other iLINCS signatures is precomputed using the extreme Pearson's correlation[74,75] of signed significances of all genes. The signed significance of the *i*th gene is defined as

$$s_i = \text{sign}(d_i) * (-\log_{10}(p_i)), for\ i = 1, \ldots, N, \tag{1}$$

and the signed significance signature is $\mathbf{s} = (s_1,...,s_N)$. The extreme signed signature $\mathbf{e} = (e_1,...,e_N)$ is then constructing by setting the signed significances of all genes other than the top 100 and bottom 100 to zero:

$$e_i = \begin{cases} s_i, & if\ s_i \geq s^{100}\ or\ s_i \leq s^{-100} \\ 0, & otherwise \end{cases} \tag{2}$$

Where $s^{100}$ is the 100th most positive $s_i$ and $s^{-100}$ is the 100th most negative $s_i$. The extreme Pearson correlation between two signatures is then calculated as the standard Pearson's correlation between the extreme signed significance signatures.

If the query signature is created from an iLINCS dataset, or directly uploaded by the user, the connectivity with all iLINCS signatures is calculated as the weighted correlation between the two vectors of log-differential expressions and the vector of weights equal to [-log10($P$ value of the query) −log10($P$ value of the iLINCS signature)][76]. When the user-uploaded signature consists of only log-differential expression levels without $P$ values, the weight for the correlation is based only on the $P$ values of the iLINCS signatures [−log10($P$ values of the iLINCS signatures)].

If the query signature uploaded by the user consists of the lists of up- and downregulated genes connectivity is calculated by assigning −1 to downregulated and +1 to upregulated genes and calculating Pearson's correlation between such vector and iLINCS signatures. The calculated statistical significance of the correlation in this case is equivalent to the $t$ test for the difference between differential expression measures of iLINCS signatures between up- and downregulated genes.

If the query signature is uploaded by the user in a form of a gene list, the connectivity with iLINCS signatures is calculated as the enrichment of highly significant differential expression levels in iLINCS signature within the submitted gene list using the Random Set analysis[77].

## Perturbagen connectivity analysis

The connectivity between a query signature and a "perturbagen" is established using the enrichment analysis of individual connectivity scores between the query signature and set of all L1000 signatures of the perturbagen (for all cell lines, time points, and concentrations). The analysis establishes whether the connectivity scores as a set are "unusually" high based on the Random Set analysis[77].

## iLINCS signature libraries

LINCS L1000 signature libraries (Consensus gene knockdown signatures (CGS), Overexpression gene signatures and Chemical perturbation signatures): for all LINCS L1000 signature libraries, the signatures are constructed by combining the Level 4, population control signature replicates from two released GEO datasets (GSE92742 and GSE70138) into the Level 5 moderated Z scores (MODZ) by calculating weighted averages as described in the primary publication for the L1000 Connectivity Map dataset[2]. For CP signatures, only signatures showing evidence of being reproducible by having the 75th quantile of pairwise spearman correlations of level 4 replicates (Broad institute distil_cc_q75 quality control metric[2]) greater than 0.2 are included. The corresponding $P$ values were calculated by comparing MODZ of each gene to zero using the Empirical Bayes weighted $t$ test with the same weights used for calculating MODZs. The shRNA and CRISPR knockdown signatures targeting the same gene were further aggregated into Consensus gene signatures (CGSes)[2] by the same procedure used to calculate MODZs and associated $P$ values.

**LINCS-targeted proteomics signatures.** Signatures of chemical perturbations assayed by the quantitative targeted mass spectrometry proteomics P100 assay measuring levels 96 phosphopeptides and GCP assay against ~60 probes that monitor combinations of post-translational modifications on histones[5].

**Disease-related signatures.** Transcriptional signatures constructed by comparing sample groups within the collection of curated public domain transcriptional dataset (GEO DataSets collection)[34]. Each signature consists of differential expressions and associated $P$ values for

all genes calculated using Empirical Bayes linear model implemented in the *limma* package.

**ENCODE transcription factor-binding signatures.** Genome-wide transcription factor (TF) binding signatures constructed by applying the TREG methodology to ENCODE ChiP-seq[78]. Each signature consists of scores and probabilities of regulation by the given TF in the specific context (cell line and treatment) for each gene in the genome.

**Connectivity map signatures.** Transcriptional signatures of perturbagen activity constructed based on the version 2 of the original Connectivity Map dataset using Affymetrix expression arrays[17]. Each signature consists of differential expressions and associated $P$ values for all genes when comparing perturbagen-treated cell lines with appropriate controls.

**DrugMatrix signatures.** Toxicogenomic signatures of over 600 different compounds[79] maintained by the National Toxicology Program[80] consisting of genome-wide differential gene expression levels and associated $P$ values.

**Transcriptional signatures from EBI Expression Atlas.** All mouse, rat and human differential expression signatures and associated $P$ values from manually curated comparisons in the Expression Atlas[8].

**Cancer therapeutics response signatures.** These signatures were created by combining transcriptional data with drug sensitivity data from the Cancer Therapeutics Response Portal (CTRP) project[81]. Signatures were created separately for each tissue/cell lineage in the dataset by comparing gene expression between the five cell lines of that lineage that were most and five that were least sensitive to a given drug area as measured by the concentration-response curve (AUC) using two-sample $t$ test.

**Pharmacogenomics transcriptional signatures.** These signatures were created by calculating differential gene expression levels and associated $P$ value between cell lines treated with anti-cancer drugs and the corresponding controls in two separate projects: The NCI Transcriptional Pharmacodynamics Workbench (NCI-TPW)[82] and the Plate-seq project dataset[4].

## Constructing signatures from iLINCS datasets

The transcriptomics or proteomics signature is constructed by comparing expression levels of two groups of samples (treatment group and baseline group) using Empirical Bayes linear model implemented in the *limma* package[83]. For the *GREIN* collection of GEO RNA-seq datasets[84], the signatures are constructed using the negative binomial generalized linear model as implemented in the *edgeR* package[85].

## Analytical tools, web applications, and web resources

Signatures analytics in iLINCS is facilitated via native R, Java, JavaScript, and Shiny applications, and via API connections to external web application and services. Brief listing of analysis and visualization tools is provided here. The overall structure of iLINCS is described in Supplemental Fig. 2.

Gene list enrichment analysis is facilitated by directly submitting lists of gene to any of the three prominent enrichment analysis web tools: Enrichr[20], DAVID[21], ToppGene[22]. The manipulation and selection of list of signature genes is facilitated via an interactive volcano plot JavaScript application.

Pathway analysis is facilitated through general-purpose enrichment tools (Enrichr, DAVID, ToppGene), the enrichment analysis of Reactome pathways via Reactome online tool[23], and internal R routines for SPIA analysis[86] of KEGG pathways and general visualization of signatures in the context of KEGG pathways using the KEGG API[24].

Network analysis is facilitated by submitting lists of genes to Genemania[25] and by internal iLINCS Shiny Signature Network Analysis (SigNetA) application.

Heatmap visualizations are facilitated by native iLINCS applications: Java-based FTreeView[87], modified version of the JavaScript-based Morpheus[88] and a Shiny-based HeatMap application, and by connection to the web application Clustergrammer[29].

Dimensionality reduction analysis (PCA and t-SNE[89]) and visualization of high-dimensional relationship via interactive 2D and 3D scatter plots is facilitated via internal iLINCS Shiny applications.

Interactive boxplots, scatter plots, GSEA plots, bar charts, and pie charts used throughout iLINCS are implemented using R ggplot[90] and plotly[91].

Additional analysis is provided by connecting to X2K Web[26] (to identify upstream regulatory networks from signature genes), L1000FWD[27] (to connect signatures with signatures constructed using the Characteristic Dimension methodology[92]), STITCH[28] (for visualization of drug-target networks), and piNET[30] (for visualization of gene-to-pathway relationships for signature genes).

Additional information about drugs, genes, and proteins are provided by links to, LINCS Data Portal[31], ScrubChem[32], PubChem[33], Harmonizome[93], GeneCards[94], and several other databases.

### Gene and protein expression dataset collections

iLINCS backend databases provide access to more than 34,000 preprocessed gene and protein expression datasets that can be used to create and analyze gene and expression protein signatures. Datasets are thematically organized into eight collections with some datasets assigned to multiple collections. User can search all datasets or browse datasets by collection.

**LINCS collection.** Datasets generated by the LINCS data and signature generation centers[7].

**TCGA collection.** Gene expression (RNASeqV2), protein expression (RPPA), and copy number variation data generated by TCGA project[63].

**GDS collection.** A curated collection of GEO Gene Datasets (GDS)[34].

**Cancer collection.** An ad hoc collection of cancer-related genomics and proteomic datasets.

**Toxicogenomics collection.** An ad hoc collection of toxicogenomics datasets.

**RPPA collection.** An ad hoc collection of proteomic datasets generated by Reverse Phase Protein Array assay[95].

**GREIN collection.** Complete collection of preprocessed human, mouse, and rat RNA-seq data in GEO provided by the GEO RNA-seq Experiments Interactive Navigator (GREIN)[84].

**Reference collection.** An ad hoc collection of important gene expression datasets.

### Reporting summary

Further information on research design is available in the Nature Research Reporting Summary linked to this article.

## Data availability

All data used in the analyses is already in the public domain. The datasets can be downloaded from iLINCS, or from the original sources which are provided on the dataset and signature landing pages in iLINCS GUI. The source LINCS L1000 datasets are GSE92742 and GSE70138, the aging rat rapamycin treatment dataset is GSE108978 and the SARS-Cov-2 infection dataset is GSE147507. The TCGA datasets were downloaded from Genomic Data Commons [https://portal.gdc.cancer.gov/] using the TCGAbiolinks R package [https://bioconductor.org/packages/release/bioc/html/TCGAbiolinks.html]. Supplemental Use Cases describe how to use iLINCS GUI to access all signatures and datasets used in the analyses.

## Code availability

All analyses in iLINCS are based on standard, previously described, and implemented statistical and bioinformatics procedures. All procedures used in iLINCS are implemented in the standard R packages and detailed Supplemental Methods. We also provide RStudio scripts that demonstrates the use of iLINCS API and the R script that provides offline implementation of various connectivity metrics (Supplemental Quality Control) used in iLINCS in the GitHub repository (https://github.com/uc-bd2k/ilincsAPI), and the compressed archive is provided in the Software Supplement.

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

## Acknowledgements

We would like acknowledge helpful comments about the manuscript provided by Dr. Albert Lee and Dr. Ajay Pillai. We would also like to acknowledge the contributions of Dr. Vineet Joshi and Dr. Mukta Phatak to the development of early prototypes of the iLINCS portal. The development of iLINCS was funded by the LINCS-BD2K Data Coordination and Integration Center U54 grant: U54HL127624 (A.M., M.M., and S.S.), Center for Environmental Genetics Bioinformatics Core: P30ES006096 (M.M.), and the Computational Tool Development and Integrative Data Analysis for LINCS U01 grant: U01HL111638 (M.M.).

## Author contributions

M.P. designed and implemented the front end user interfaces and backend the API infrastructure; M.K. designed the software infrastructure including docker containers and the docker swarm; B.S. designed and implemented portions of the user interfaces, visualizations help pages, and performed version updates and maintenance of the whole system; M.F.N. and M.M. wrote the analytical engine R code; J.M., J.V., and J.F.R. tested and contributed to development of the user interfaces; W.N. created, maintained and populated backend databases; L.Z. constructed LINCS L1000 signatures; M.F.N. performed connectivity map analysis between all precomputed signatures; S.E.D. preprocessed toxicogenomics datasets; K.R. designed and maintained hardware infrastructure; M.P., M.F.N., B.S., N.M., S.W., R.K., H.X., J.B., and M.F.B. designed and implemented specific visualization tools; V.S., A.K., D.V., and S.C.S. contributed metadata for LICNS perturbation signatures and facilitated integration with LINCS Data Portal; D.J.B.C. and A.M. facilitated integration with bioinformatics tools developed by the Ma'a-yan Laboratory; J.M. and M.M. designed use cases and performed the analysis; M.M. conceived the project, lead the development, created figures and wrote the manuscript with contributions from other authors.

## Competing interests

The authors declare no competing interests.
