## [Peer Review File · Nature Communications]

Reviewers' Comments:

Reviewer #1:

Remarks to the Author:

The authors presented a tool, termed iLINCS, in which they assembled large-scale, publicly available gene/protein expression data, and various published bioinformatics and systems biology tools. Although the iLINCS contained large-scale, publicly available gene/protein expression datasets, there are several major concerns in current version. The current iLINCS framework is lack of novelty and the authors are suggested to highlight novelty compared to various published bioinformatics and systems biology tools they integrated. Second, the authors only illustrated 3 case studies by focusing on cancer (like breast cancer) as those major cancer types have more published datasets. The overall performance of the iLINCS is not evaluated for other diseases, including rare cancer types and others. Various confounding factor adjustment, including sex, age, tumor stages, bench effect, were missed in differential expression analysis. Several issues are needed attention below, which may improve the manuscript further.

1. The authors have assembled large-scale, heterogenous gene expression datasets from different resources; Yet, many key factors during differential expression analysis were missed in current Methods or supplemental Methods. For example, how the authors adjusted various confounding factors, including sex, age, tumor stages, bench effect.
2. The number of differentially expressed genes are highly dependent with the sample size. How the users can evaluate the reproducibility of gene signatures from the iLINCS workflow?
3. The iLINCS workflow assemble many publicly available gene/protein expression datasets and published bioinformatics and systems biology. One big concern is the reproducibility of heterogenous gene/protein expression datasets and published bioinformatics and systems biology as each dataset or tool (bioinformatics and systems biology) has different data noise and experimental/computational errors.
4. The authors integrate gene expression data across various tissues and cell lines, especially for cancer type. If the authors obtain inconsistent expression patterns between tissue-based and cell-based gene expression, how the iLINCS workflow deal with these issues. The authors are suggested to illustrate correlation of gene expression between tissues and cell lines.
5. The authors only evaluated the performance of the iLINCS workflow using 3 case studies. The users will be more interesting to see the overall performance of the iLINCS workflow across different disease area, including disease mechanism, drug predictions (including pharmacogenomics and drug repurposing as well). The authors are suggested to provide more solid validation for each module to illustrate accuracy and the reproducibility of the iLINCS workflow for users.
6. It looks like that the iLINCS workflow centered for cancer research communities. If yes, the title should be confused on cancer. How about other disease area and tissue-specific disease-gene expression signatures be integrated in the the iLINCS workflow as well.
7. It is unclear how the iLINCS workflow offer cell type-specific, tissue-specific, or cell type-tissue matched gene and protein signature analysis for a specific disease or a cancer type.
8. There are many functional modules for iLINCS interfaces, which requires that the users must have specific bioinformatics expertise and knowledge by using iLINCS. These complex functional modules reduce the enthusiasm of the proposed iLINCS for general users.

Reviewer #2:

Remarks to the Author:

This paper describes a set of R scripts that can be used to integrate multiple omics datasets and mine new and pre-existing omics data for signatures characteristic of disease or drug perturbations. The analysis workflows include visualization tools (volcanic plots, etc), differential gene expression and pathway analysis (linked to multiple enrichment analysis tools and pathways).

One strength of the paper is the extensive databases the software can mine: >10,000 processed omics datasets, >220,000 omics signatures and >109 statistically significant "connections" between signatures. The data include transcriptomic and proteomic datasets. To demonstrate the

utility of their scripts the authors present three use case examples.

The authors make their data and scripts available as R Notebooks. However, when I tried to run them, I couldn't. (generated too many errors)

Also, the claim that the iLINCS platform does not require programming skills is an exaggeration. Although the R scripts make the life of the data analysts easier, the user still needs to have quite a bit of programming experience to troubleshoot.

A couple of other drawbacks:

1. These scripts depend on existing packages and need to be updated each time one of these packages changes.
2. Many disease, especially chronic diseases, are syndromes, consisting of multiple subtypes. How does this affect the iLINC analysis platform? There is a large body of work that looking of disease signatures, which the authors do not seem to acknowledge. (for example, Bigler et al, 2016; Buschur et al, 2020)
3. LINCS, which is part of their datasets, only contains a small number of transcripts, which limits the usefulness of this software. also the implications of this fact to the perturbation signature identification need to be discussed.

Overall the authors did a very good job putting together this resource and it will undoubtedly be useful to some researchers. But probably not as many as the authors think.

Reviewer #3:

Remarks to the Author:

Summary

Pilarczyk et al. present a web-based platform and API to query gene signatures of cellular perturbations directly or indirectly related to user provided gene lists, or to retrieve transcriptomics and proteomics data sets such as LINCS L1000 and EBI Expression Atlas sitting in the backend database of LINCS and perform further analysis on them. An ensemble of bioinformatics suites is provided to support various modes of data analysis through the portal, enabling biologists not familiar with scripting-based data analysis to explore data at will. Overall, the portal appears to serve as a nice gateway for exploring the vast gene expression database LINCS team have curated over the years.

Major comments

- Having said that, it is my general impression that the current article does not live up to the hype that the data resource and the portal offer. This is mainly because the brief case studies merely scratch the surface in showcasing key functionalities of the tool, and perhaps make readers wonder what the biological question is at hand by the time they reach the end of an example case. In that regard, the case study portion of the paper can be vastly expanded, improved, or even re-designed with more exciting examples. One possibility here is to have a single consistent data example that proceeds in multiple stages, with a newly emerging therapeutic agent, instead of having three cases studies that are more or less disconnected. Just a suggestion.
- In use case 2, the key story here is that the transcriptional response signature of a given therapeutic agent can be related to the pathways enriched in pre-compiled LINCS signatures and the "relational" analysis can recover the underlying mechanistic routes and direct targets that are not necessarily affected at the mRNA transcript level. But one can question this potential benefit by asking how many other pathways appear as significantly enriched as a result of connecting to the other LINCS CGSes and their transcriptional signatures. Wouldn't the expansion of gene lists almost always increase the list of related pathways enriched in the associated transcriptional signatures? How do we comb through increasing amount of reported information and prevent false positives in this process?

- In use case 3, page 12, it was very difficult to understand the authors interpretation, Figure 3, and Supplemental Table 3, all together. It is way too descriptive to follow. Are the authors suggesting new regimen of the aforementioned inhibitors will alter the tumor cell identity of Her2-enriched invasive tumors? Generally speaking, my problem with these examples is that, all of a sudden, we have to understand the mRNA and RPPA data in a sub-cohort of TCGA's BRCA (excluding basal-like subtype, for instance) with the lens of pharmacological intervention reflected in the inhibitor-treated cell line data in the LINCS database. Will it be possible to make the objectives of each case study up front, and then include a concise summary statement at the end?

Minor comments

- Page 3 of Supplemental Methods, in the last equation, the second d-bar should have a superscript "L".
- Parts of Supplemental Results can be moved to the main text, which will make things look more concrete and easier to follow. I strongly urge the authors to redistribute the materials within the word limit. For example, they can move some of the lengthy enumeration of tools implemented in iLINCS from the main text to Supplemental Methods.
- Page 4 of the main text, in the first line, "lead to"  "leading to"?
- Page 4 of the main text, in the third last line, "network analyses approaches"  "network analysis approaches"?
- Have you considered incorporating CPTAC proteomics data into the backend? Of course, these data are in much smaller sample sizes due to the limited throughput of MS-based proteomics, but it has novel components such as phosphoproteomics that RPPA does not offer with a sufficiently wide coverage of the proteome. There may be some value in including their data.
- Once again, I would encourage the authors to think about the possibility of spurious association propagating through signature similarity-based analysis, or at least discuss this aspect in the Discussion.

Response to reviewer's comments

We appreciate very much the thorough review and thoughtful comments provided by reviewers. In the revised manuscript we went to a great length to make sure that we address all critiques and suggestions. The revised use cases now provide more in-depth analyses and better demonstrate the breath of the applications and analyses enabled by iLINCS. For example, we added a new use case describing the use of iLINCS in dissecting the mechanisms of SARS-Cov2 infection and the use of signature connectivity approach to nominate putative therapeutics that modulate the effects of the infection. Furthermore, revised use cases now directly address the mechanisms for dealing with confounding factors and include the discussion of the limitations of data, signatures and methods currently implemented in iLINCS. Finally, we curated and analyzed the content of published studies already referencing the use of iLINCS and demonstrate a wide-ranging use in terms of diseases studied with iLINCS.

We also modified iLINCS user interfaces, re-structured on-line help, and added 23 links to context-specific video tutorials which illustrate all major workflows, use cases presented in the paper and other aspects of iLINCS functionality. We would like to emphasize again that the primary avenue for interacting with iLINCS is its web graphical user interface (GUI). This type of use and execution of all use cases require only a web browser and a mouse, and does not require any programming. For users with adequate programming skills, we also provide “programmable access” to iLINCS via APIs. For these users, we provide R notebooks that re-capitulate the results obtained via GUI, and feature capabilities of the iLINCS API. However, the use of these notebooks requires some programming as well as system administration skills for installing all required software and R packages. We have updated the GitHub (<https://github.com/uc-bd2k/ilincsAPI>) repository that maintains the notebooks to provide detailed information about the system requirements and packages that need to be installed to execute scripts in notebooks, and provide instructions for installing the packages. For computationally savvy users, we also provide Dockerfiles for easy creation of Docker containers capable of executing all scripts in the GitHub repository. Since the Gigantum system seized operations in the meantime, we created a Jupyter R notebook demonstrating the use of iLINCS API that can be executed on the Google Collaborative platform without any additional programming. The Jupyter notebook and the link to it within the Google Collaborative is included in the GitHub repository.

In the revised manuscript, we also address the future directions in the development of iLINCS and the design features that will facilitate the continual updates of back-end data and signatures. For example, since the initial submission, the number of pre-processed datasets has more than tripled to >34,000. The use of iLINCS by the community also continued to increase since the initial submission and there are currently on average >200 unique daily visitors, excluding automated servers and trivial visits. Our curated literature search identified 118 peer reviewed publications referencing the use of iLINCS within last three years. In the answer to Reviewers' comments, we have also included a discussion of ongoing development and future extensions. Altogether, iLINCS is a vibrant, ongoing project, that already has a well-established user community which we hope will increase significantly after this manuscript is published.

Below we provide point-by-point response and the details of how their comments have been addressed in the revised manuscript.

Reviewer #1 (Expertise: systems biology, clinical data integration):

The authors presented a tool, termed iLINCS, in which they assembled large-scale, publicly available gene/protein expression data, and various published bioinformatics and systems biology tools. Although the iLINCS contained large-scale, publicly available gene/protein expression datasets, there are several major concerns in current version. The current iLINCS framework is lack of novelty and the authors are suggested to highlight novelty compared to various published bioinformatics and systems biology tools they integrated. Second, the authors only illustrated 3 case studies by focusing on cancer (like breast cancer) as those major cancer types have more published datasets. *The overall performance of the iLINCS is not evaluated for other diseases, including rare cancer types and others. Various confounding factor adjustment, including sex, age, tumor stages, bench effect, were missed in differential expression analysis.* Several issues are needed attention below, which may improve the manuscript further.

Answer: We appreciate Reviewer's concerns about the scope and utility of iLINCS, and mechanisms for confounding factor adjustments, and address them in detail in answers to specific issues listed below. The main novelty of the iLINCS is in removing roadblocks for studying disease mechanisms and pharmacological interventions in the context of the "connectivity map" (CMAP) and related analyses. There is no other user-friendly tool that comes near in terms of facilitating various aspects of the CMAP analysis in terms of the scope of the analyses and the data that can be used in these analyses. The canonical iLINCS workflow can be summarized as: 1) Select an RNA-seq dataset in GEO; 2) Perform differential gene expression analysis to construct a transcriptional signature; 3) Perform signature connectivity map (CMAP) analysis against LINCS perturbation signatures as well as other sets of transcriptional signatures; 4) Perform pathway analysis of genes perturbed by genetic or chemical perturbation giving rise to "connected signatures". There is no other user-friendly bioinformatics analysis tool that can execute this workflow. Expert data scientists with conceptual understanding of the analysis and coding skills would be, with significant effort, be able to use existing analytical packages and databases to implement and execute such a workflow, and other workflows demonstrated in our use cases. With iLINCS however, these analyses can be completed in a few minutes without any programming. We have introduced this discussion point to clarify where the novelty of iLINCS comes from. In particular, the direct application of LINCS perturbation signatures to circumvent the inadequacies of standard pathway enrichment analyses (Use case 1) has been discussed in our previous work, but it is still an innovative analytical approach, and iLINCS is the only bioinformatics tool that implements it.

1. The authors have assembled large-scale, heterogenous gene expression datasets from different resources; Yet, many key factors during differential expression analysis were

missed in current Methods or supplemental Methods. For example, how the authors adjusted various confounding factors, including sex, age, tumor stages, bench effect.

Answer: The primary mechanism for dealing with various confounding factors implemented in iLINCS is filtering by levels of possible confounding factors. This is the strategy applied by the vast majority of published studies to date. In the revised manuscript, we explicitly demonstrate this functionality in two analyses: 1) The analysis of SARS Cov 2 infected cells, where the comparison between infected and control cells is limited to a specific A549 cell line to factor out differences in response in different variants of the A549 (Use case 3 in the revised manuscript); and 2) The analysis of rapamycin treatment in aging rat livers (Use case 1 in the revised manuscript) where the potentially confounding effects of age and target organ were factored out by filtering samples and focusing only on samples from liver and 24 months old rats. The rapamycin signature was constructed by comparing expression profiles of livers in rapamycin treated rats to the vehicle control at 24 months of age (Fig 3 heatmap).

Multivariate regression type of models are not common in analyses of omics data. That said, iLINCS also facilitates one co-factor adjustment by fitting a two-factor negative binomial generalized linear model for GEO RNA-seq data. This is useful in situations when samples can be matched by a higher-level experimental unit. For example, comparing changes in expression levels from a baseline between two types of samples instead of direct comparison of expression levels between two groups. The same mechanism can be used to adjust for batch effects if experimental metadata includes the sample property that defines batch levels.

It should be noted that more specialized batch adjustment tools are not made directly available in iLINCS. Detecting batch effects and assessing the effectiveness of various adjustment strategies is a complex task, requiring careful assessment of the sources of technical vs. biological variability. In situations when this is necessary, iLINCS still simplifies significantly the re-use of public domain data. For example, our RNA-seq pipeline systematically downloads the datasets from the Short Read Archive, quantifies the expression levels of all genes and creates the sample metadata. User can then download pre-processed analysis-ready dataset for off-line analysis and skip rather arduous processing of raw sequencing data.

2. The number of differentially expressed genes are highly dependent with the sample size. How the users can evaluate the reproducibility of gene signatures from the iLINCS workflow?

Answer: iLINCS implements the standard procedures for evaluating the reproducibility in differential gene expression analysis by providing p-values of differential expressions as the measure of statistical significance. Everything else being equal (underlying true magnitude of difference and the biological variability), comparisons with more samples will generate more statistically significantly differentially expressed genes. This will always be the case. However, since iLINCS also provides the magnitude of the differential expression, and users themselves can use both of these measures (significance and effect

size or magnitude) to select significant genes under the assumption that the larger magnitude of change may correspond to biologically stronger signal. This is a standard practice in transcriptomic data analysis.

The selection of differentially expressed genes in iLINCS is facilitated via interactive volcano plot tool for selecting differentially expressed genes based on the combination of the p-value and log₂-ratio cut-offs. In the revised manuscript, we included examples of using the volcano tool (new Figures 5C and 7A) to select genes for downstream analysis and the use of the tools is demonstrated in the corresponding written and video tutorials.

It is important to emphasize that iLINCS implements current state-of-the art methodologies for differential gene and protein expression analysis, and default iLINCS workflows can reproduce the analyses in the primary publications describing the most of the datasets. The experimental reproducibility and generalizability of the results will depend on additional experimental design factors which cannot be readily assessed computationally, but require scrutinizing the description of the experimental design. iLINCS aims to remove roadblocks from re-analysis of omics datasets and provides default choices that will in most situations result in the correct analysis. However, it does not obviate the need for understanding the nature of the dataset used and the conceptual understanding of the analyses performed for understanding and interpreting results it provided.

3. The iLINCS workflow assemble many publicly available gene/protein expression datasets and published bioinformatics and systems biology. One big concern is the reproducibility of heterogenous gene/protein expression datasets and published bioinformatics and systems biology as each dataset or tool (bioinformatics and systems biology) has different data noise and experimental/computational errors.

Answer: This certainly is a problem at the current stage of omics data use in biomedical sciences in general. iLINCS provides access to a wide range of datasets and state of the art tools to analyze them. In Quality Control Supplement, we demonstrate that the data in iLINCS databases accurately represents the data in the in the public domain repositories from which the data is obtained. We also demonstrate that the iLINCS analytical engine accurately performs the analyses that it is designed to perform. In terms of specialized bioinformatics tools linked in iLINCS for performing pathway and network analysis and for visualizing data, we use state of the art, highly cited tools that are widely accepted by the community. For pathway analysis, iLINCS connects to four such highly used resources (Enrichr, TopGene, DAVID and Reactome) and user can easily perform all four analyses and scrutinize the consistency of the results. We demonstrate this in the Supplemental Workflow S1 and the corresponding video tutorial.

The statistical noise in data and signatures is assessed by providing measures of statistical significance (p-values) in all pre-computed iLINCS signatures and signatures de-novo created within iLINCS. The iLINCS default connectivity map metric (weighted Person's correlation coefficient, described in Supplemental Methods) uses both differential expression and inherent signal to noise ratio as quantified by the measure of statistical

significance (p-value) to identify connected perturbation signatures. In this sense, iLINCS automatically accounts for the level of noise in the query signature.

In summary, iLINCS accurately represents primary data, provides widely used, state-of-the-art analytics that accounts for the observed data variability when possible, and reports measures of statistical significance. It provides consistent and rigorous analysis results using the state-of-the-art analysis protocols. At the same time, it allows analyses that could take a bioinformatician days to orchestrate using off line analysis platforms to be completed in only a few minutes, without any programming, as demonstrated by our workflows.

4. The authors integrate gene expression data across various tissues and cell lines, especially for cancer type. If the authors obtain inconsistent expression patterns between tissue-based and cell-based gene expression, how the iLINCS workflow deal with these issues. The authors are suggested to illustrate correlation of gene expression between tissues and cell lines.

Answer: There are many biological reasons for possible differences in transcriptomic signatures obtained from tissues and cell lines in general. iLINCS provides means for assessing such discrepancies and facilitates in-depth analysis of genes that drive the similarities between tissue and cell line signatures as well as genes that show inconsistent behavior.

As Reviewer suggested, we illustrated the correlation between signatures in tissues and cell lines by extending our mTOR signaling use case to verify that mTOR signaling modulation is also detectable in complex tissues. We used iLINCS to re-analyze the aging study dataset of the effect of rapamycin in aged rat livers (GEO datasets GSE108978). The rapamycin signature was constructed by comparing expression profiles of livers in rapamycin treated rats to the vehicle control at 24 months of age (New Fig 3 heatmap). The signature correlated strongly with CP signatures of chemical of chemicals targeting mTOR pathway genes (New Fig 3 bar plot).

5. The authors only evaluated the performance of the iLINCS workflow using 3 case studies. The users will be more interesting to see the overall performance of the iLINCS workflow across different disease area, including disease mechanism, drug predictions (including pharmacogenomics and drug repurposing as well). The authors are suggested to provide more solid validation for each module to illustrate accuracy and the reproducibility of the iLINCS workflow for users.

Answer: We provide a quality control evaluation of iLINCS data and analysis in the Supplemental Quality Control (SQC) document. In this document we demonstrate the accuracy and consistency of iLINCS signatures, datasets and analytical procedures. These results indicate that given the methodology used and data set deposited, iLINCS will perform the analysis correctly. Here we briefly summarize the quality control (QC) analyses.

The quality control (QC) of iLINCS signatures is performed by establishing that publicly released L1000 signatures are accurately represented in iLINCS databases. The consistency and reproducibility of signatures is addressed by a comprehensive self-connectivity analysis of signatures generated by the same perturbation. The consistency of the signatures is also tested by performing “self-connectivity” analysis of iLINCS signatures using the Broad clue.io platform (Quality Control of Signature Libraries in SQC document).

The accuracy and consistency of iLINCS datasets was assessed by comparing iLINCS datasets to results obtained by another group’s independent processing efforts. The QC of microarray GEO GDS datasets data and metadata was performed by comparing them to EBI Expression Atlas and RNA-seq datasets were compared to the recount2 project datasets platform (Quality Control of Omics Datasets in SQC document).

The accuracy of iLINCS computational results was tested by a comprehensive off-line re-analysis. The key analyses steps tested involved: Differential expression analysis used in constructing the signatures and the connectivity analysis using different measures of connectivity implemented in iLINCS (The Accuracy and the Consistence of iLINCS Analysis Results in SQC document). We share the complete set of R scripts used for off-line analysis and comparisons in our publicly accessible GitHub repository to be scrutinized and used by bioinformaticians for their own independent tests.

In terms of the conceptual utility of the analyses facilitated by iLINCS across different diseases and research areas, this is evident by the past and current usage of these types of analyses across a wide range of human diseases. The validity or the utility of re-using omics datasets in public domain in general, seems self-evident. Thousands of high impact papers have demonstrated that such analyses can yield incredibly useful results. The utility of the CMAP analyses at this point is also evidenced by the referenced usage of CMAP data in the peer reviewed publication. The original Science 2006, CMAP paper has been referenced more than 4,000 times and the most recent Cell 2017 paper describing L1000 dataset has been referenced more than 1,000 times. We collected and analyzed the disease topics studied in the publications referencing the CMAP papers using the Dimensions platform (<https://www.dimensions.ai/>) (Figures in the answer to issue 6) which indicates the very wide usage of CMAP across a wide range disease.

6. It looks like that the iLINCS workflow centered for cancer research communities. If yes, the title should be confused on cancer. How about other disease area and tissue-specific disease-gene expression signatures be integrated in the iLINCS workflow as well.

Answer: We agree that our initial use cases were to some extend cancer centric. However, iLINCS datasets cover a wide range of diseases, and CMAP analysis in general and iLICNS in particular have been applied in studies of a wide range of diseases. We addressed this in the revised manuscript by restructuring the use case 1 and place the mTOR regulation in the wider context, by adding a new Covid-19 related use case and by providing the summary of the iLINCS datasets and applications to date by disease type.

Our canonical example of mTOR signaling dysregulation and modulation is not cancer specific. mTOR signaling dysregulation is involved in numerous human diseases and these examples are applicable in diseases beyond cancer. In the revised manuscript we re-wrote this section of the results to emphasize the wide range of human diseases that are associated with mTOR pathway dysregulation. We also added a new use case related to COVID 19 drug repositioning. Altogether, we believe that the revised manuscript is much less cancer-specific and represents better the iLINCS scope.

Supplemental Figure 3. Top 8 disease categories associated with publications referencing the two primary CMAP papers (CMAP references) and publications referencing iLINCS.

In the revised manuscript, we also provide summaries of disease-related annotations of iLINCS datasets that clearly show iLINCS is applicable in a wide range of disease categories revealing thousands of datasets analyzable through iLINCS are not cancer-specific (Table in the Revised Figure 1A).

We also collected, curated and analyzed the disease-related topics of the studies referencing the CMAP papers and iLINCS itself using the Dimensions platform (<https://www.dimensions.ai/>) (Supplemental Figure 3, and the figure on the previous). The results indicate a broad utility of the CMAP approach in general and iLINCS in particular in terms of the diseases studied. The cancers are the most studied disease category, but still represent only about a third of all use CMAP cases and even a smaller fraction of 114 peer reviewed publications referencing the use of iLINCS used in this analysis.

Finally, we referenced and highlighted three peer reviewed studies that applied iLINCS in studying diseases other than cancer:

Wu, P. et al. Integrating gene expression and clinical data to identify drug repurposing candidates for hyperlipidemia and hypertension. Nature Communications 13, 46, doi:10.1038/s41467-021-27751-1 (2022)

O'Donovan, S. M. et al. Identification of candidate repurposable drugs to combat COVID-19 using a signature-based approach. *Scientific Reports* 11, 4495, doi:10.1038/s41598-021-84044-9 (2021)

Sullivan, C. R. et al. Connectivity Analyses of Bioenergetic Changes in Schizophrenia: Identification of Novel Treatments. *Molecular Neurobiology* 56, 4492-4517, doi:10.1007/s12035-018-1390-4 (2018)

7. It is unclear how the iLINCS workflow offer cell type-specific, tissue-specific, or cell type-tissue matched gene and protein signature analysis for a specific disease or a cancer type. be integrated in the iLINCS workflow as well.

Answer: In terms of signature creation, iLINCS provides mechanisms of sub-setting of samples by any property documented in the metadata. For example, the new COVID 19 use case in the revised manuscript limits the data to only A549 cell line over-expressing the ACE2 genes before making comparison between the SARS-CoV-2 infected and mock-infected cells. As long as different cell/tissue types are annotated by experimental metadata in the primary dataset, iLINCS facilitates cell type and tissue type specific analyses.

In terms of cell and tissue type specificity of connected perturbation signatures, we modified the iLINCS user interface to display the tissue of origin of cell lines used for the signature generation. We demonstrate this utility in COVID 19 use case, where user can focus on the iLINCS signatures generated in A549 cell line, or, for example, all lung cell lines.

In terms of integrative analysis of matched transcriptomic and proteomic data, iLINCS facilitates the integration described in Use Case 2. The results of one omics dataset informs the set of genes/proteins analyzed in the other dataset. However, analyses of transcriptomic and proteomic datasets are focused on a single dataset at a time and there is no other mechanism of direct multi-omics integration beyond the post-hoc integrative analysis that we demonstrate in Use Case 2. This is a limitation of the current functionality and is one of the possible future extensions of the iLINCS functionality that we are considering. In the revised manuscript, this limitation is discussed in the Use Case 2 results and the future developments in the Discussion section.

8. There are many functional modules for iLINCS interfaces, which requires that the users must have specific bioinformatics expertise and knowledge by using iLINCS. These complex functional modules reduce the enthusiasm of the proposed iLINCS for general users.

Answer: iLINCS provides access to many different task-specific tools and we can see that this can become overwhelming for some users. In response to this concern, we revised the iLINCS user interface to better organize different types of signature follow-up analyses that can be performed for a created signature. The analysis tools are now organized in three categories: Pathway Analysis, Network Analysis, and Visualization.

To help users navigate the user interface, we also revised the iLINCS by adding context specific instructional videos as well as video tutorials for executing use-cases presented in

the paper. We also provide video for quick iLINCS introduction and a set of analysis tutorials that emphasize the bare bone analysis workflows that quickly get to the main point of the analysis without being sidetracked by optional analyses along the way.

Reviewer #2 (Expertise: computational systems biology, OMICs integration):

This paper describes a set of R scripts that can be used to integrate multiple omics datasets and mine new and pre-existing omics data for signatures characteristic of disease or drug perturbations. The analysis workflows include visualization tools (volcanic plots, etc), differential gene expression and pathway analysis (linked to multiple enrichment analysis tools and pathways).

Answer: The iLINCS paper describes much more than a set of R scripts. R scripts that we provide for demonstrating the utility of iLINCS Application Programming Interface (API) represent only a small fraction of the iLINCS utility. The main feature of iLINCS is the streamlined graphical user interfaces (GUI) implemented in the freely accessible web application for interactive analysis of datasets and signatures. All use cases presented in the paper have been generated using iLINCS GUI. The API provides programmatic access to iLINCS functionality and extends the user base to data scientists with programming skills. In the revised manuscript we made this point more explicit to avoid the confusion.

One strength of the paper is the extensive databases the software can mine: >10,000 processed omics datasets, >220,000 omics signatures and >109 statistically significant “connections” between signatures. The data include transcriptomic and proteomic datasets. To demonstrate the utility of their scripts the authors present three use case examples.

Answer: We appreciate that Reviewer finds the access to the wide range data and signatures a strength. However, we would again like to point out that all use cases have been created using the iLINCS GUI. The step-by-step instructions are provided as Supplements and instructional videos are available online. We also provide the scripts for re-creating the use cases programmatically for computationally savvy users. In the revised manuscript we state this now clearly in the paragraph preceding the Use case 1 section.

The authors make their data and scripts available as R Notebooks. However, when I tried to run them, I couldn't. (generated too many errors)

Also, the claim that the iLINCS platform does not require programming skills is an exaggeration. Although the R scripts make the life of the data analysts easier, the user still needs to have quite a bit of programming experience to troubleshoot.

Answer: We agree with the Reviewer that the use of provided R scripts, unlike iLINCS GUI, requires programming experience. However, the primary avenue for interacting with iLINCS is its GUI and the use of iLINCS GUI does not require any programming skills. In the Supplementary materials we provide step-by-step instructions for how to re-create all use cases via iLINCS GUI without any need for programming.

The scripts we provided use additional packages, but they are included for the convenience only. iLINCS API does not depend on any of the additional packages and users have the option to wrap the API calls within python, java, javascript, or any other programming platform. The use of API comes with very much universal caveats requiring data scientists with programming and basic system administration skills.

In the original submission, we provided the code as a Gigantum cloud project. The scripts have been tested by us and by Gigantum admins and executed without any issues. We are assuming that the Reviewer ran into issues with executing the scripts from our GitHub repository on their local system. To address this, we have revised extensively the GitHub repository to provide all information about R packages that scripts use. We also include instructions on how to configure a new R system that can execute all notebooks. For users with expertise in using Docker containers, we also include Dockerfiles that can be used to create containers capable of executing the provided scripts.

Since the Gigantum system seized operations since the original submission, we also provide a Jupyter notebook “usingIlinCSApis.ipynb” that executes the complete set of iLINCS API calls used in the three use cases, and the link to the Google Collaborative platform where the script can be executed. This avenue allows users to execute iLINCS API demo using only a web browser without a need to install R locally.

A couple of other drawbacks:

1. These scripts depend on existing packages and need to be updated each time one of these packages changes.

Answer: As mentioned above, we have revised the ilincsAPI GitHub repository to clearly document the package dependencies of the R notebooks. We also include the step-by-step instruction on how to set up the R environment and install all dependences. We also provide the link for execution of the Jupyter notebook on the Google collaborative platform which requires only a web browser.

2. Many disease, especially chronic diseases, are syndromes, consisting of multiple subtypes. How does this affect the iLINC analysis platform? There is a large body of work that looking of disease signatures, which the authors do not seem to acknowledge. (for example, Bigler et al, 2016; Buschur et al, 2020)

Answer: We completely agree with the reviewer that distinguishing between different subtypes is essential for providing valuable biological insight in general. It is also essential

for creating effective signatures for CMAP analysis. In the revised manuscript we made this point by revising the write up of the breast cancer use case. There we demonstrate that contrasting two different subtypes of the breast cancer is more effective in detecting disease driver mechanisms than comparisons of the subtypes to control samples. Failure to distinguish between the two subtypes leads to even less informative signatures. These examples demonstrate the mechanisms provided by iLINCS to leverage the disease subtypes to construct informative disease signatures when subtype information is included in sample metadata. In the revised manuscript we also include the discussion of strategies for construction of more precise disease signatures, future extension that will facilitate de novo subtype creation as well as strategies for constructing informative disease signatures from single cell data (fourth paragraph in Discussion).

3. LINCS, which is part of their datasets, only contains a small number of transcripts, which limits the usefulness of this software. also the implications of this fact to the perturbation signature identification need to be discussed.

Answer: We agree that the reduced representation philosophy of LINCS signatures introduces some limitations on the utility of the LINCS L1000 datasets, but there is an overwhelming preponderance of evidence that the dataset is still very effective for its primary objective, which is CMAP analysis. This point was also clearly demonstrated in the primary Cell paper that describes the L1000 assay and data. We added the discussion of this point in the revised paper, as requested by the Reviewer, in the second to last paragraph in Use case 1. We also demonstrate that our mTOR signaling pathway enrichment analysis results are not an artifact of the L1000 data idiosyncrasies by repeating the analysis in Use case 1 with a whole-genome signature of sirolimus (new Supplemental Results 4). These results very much recapitulated the results obtained by analyzing L1000 signature of everolimus.

Overall the authors did a very good job putting together this resource and it will undoubtedly be useful to some researchers. But probably not as many as the authors think.

Answer: We are gratified by the overall positive view of iLINCS as a resource for some researchers. We also believe that thousands of returning users at this early stage provides additional evidence that iLINCS will be considered valuable to many scientists. The curated search of Google Scholar, Pubmed Central and the Dimensions databases identified 118 peer reviewed studies referencing iLINCS in last three years. We also expect that the adoption will accelerate once the paper is published

Reviewer #3 (Expertise: Computational and systems biology, OMICS integration):

Summary

Pilarczyk et al. present a web-based platform and API to query gene signatures of cellular perturbations directly or indirectly related to user provided gene lists, or to retrieve transcriptomics and proteomics data sets such as LINCS L1000 and EBI Expression Atlas sitting in the backend database of LINCS and perform further analysis on them. An ensemble of bioinformatics suites is provided to support various modes of data analysis through the portal, enabling biologists not familiar with scripting-based data analysis to explore data at will. Overall, the portal appears to serve as a nice gateway for exploring the vast gene expression database LINCS team have curated over the years.

Major comments

- Having said that, it is my general impression that the current article does not live up to the hype that the data resource and the portal offer. This is mainly because the brief case studies merely scratch the surface in showcasing key functionalities of the tool, and perhaps make readers wonder what the biological question is at hand by the time they reach the end of an example case. In that regard, the case study portion of the paper can be vastly expanded, improved, or even re-designed with more exciting examples. One possibility here is to have a single consistent data example that proceeds in multiple stages, with a newly emerging therapeutic agent, instead of having three cases studies that are more or less disconnected. Just a suggestion.

Answer: We agree with the reviewer that examples provided only scratch the surface of what is possible. The objective of our use cases was to re-create known facts using the canonical examples and demonstrate that iLINCS provides credible insight with little effort. The complexity and the scope of opportunities is vast. In the revised manuscript, we added a COVID 19 use case. We also revised the way how the use cases are presented to provide the depth and the conceptual integration across the use cases.

- In use case 2, the key story here is that the transcriptional response signature of a given therapeutic agent can be related to the pathways enriched in pre-compiled LINCS signatures and the “relational” analysis can recover the underlying mechanistic routes and direct targets that are not necessarily affected at the mRNA transcript level. But one can question this potential benefit by asking how many other pathways appear as significantly enriched as a result of connecting to the other LINCS CGSes and their transcriptional signatures. Wouldn't the expansion of gene lists almost always increase the list of related pathways enriched in the associated transcriptional signatures? How do we comb through increasing amount of reported information and prevent false positives in this process?

Answer:

The “relational” pathway analysis shares methodological shortcomings with the standard enrichment/pathway analyses of lists of differentially expressed genes, such as, for example, overlapping pathways. While indeed other pathways do show up as significant in our use case, the MTOR pathway shows the strongest association as measured by the

level of statistical significance and the magnitude of enrichment. One can then follow up a standard practice in the analysis of differentially expressed genes of examining relationships between enriched pathways in terms of overlapping gene, which is facilitated by Enrichr, for instance. We can “drill in” into the results and examine the nature of the enrichment in other pathways and the genes driving this enrichment. A closer examination of most significantly enriched pathways in Use case 1 reveals that they all contain PI3K-AKT-MTOR component as a sub-pathway, and many of the genes that drive the associations with other four most enriched pathways are common with the mTOR pathway as shown in new Figure 3 (on the left) where yellow squares indicate membership of genes (rows) in enriched pathways (columns) illustrates this point.

In principle, adding genes to the list will not necessarily implicate the additional pathways since the enrichment level is measured as the ratio of the number of genes that belong to the pathway and the overall size of the list. The association with the pathway would strengthen only if added genes are disproportionately associated with the pathway. That said, an increase in the number of “significant” genes does lead to increased power of detecting faint associations as statistically significant. It is difficult to define an algorithmic approach to selecting “meaningful” associations, but the most commonly used strategies are reflected in our analysis: considering most significant associations and scrutinizing the gene overlaps in “significant” pathways.

In use case 3, page 12, it was very difficult to understand the authors interpretation, Figure 3, and Supplemental Table 3, all together. It is way too descriptive to follow. Are the authors suggesting new regimen of the aforementioned inhibitors will alter the tumor cell identity of Her2-enriched invasive tumors? Generally speaking, my problem with these examples is that, all of a sudden, we have to understand the mRNA and RPPA data in a sub-cohort of TCGA’s BRCA (excluding basal-like subtype, for instance) with the lens of pharmacological intervention reflected in the inhibitor-treated cell line data in the LINCS database. Will it be possible to make the objectives of each case study up front, and then include a concise summary statement at the end?

Answer: We revised extensively this use-case to provide the context and explain why comparing directly two cancer subtypes makes sense in the context of identifying the pharmacological intervention. We also motivate the use of both RPPA and mRNA data up-front before diving into the analyses.

The objective of our analysis is to identify molecular features of the two cancer subtypes that can be used as the targets of pharmacological intervention. In this, we would like to identify the core driving events and tune out the general features of cancer that are

common to most cancers. To do that, it is opportune to match, as much as possible, the two sample types used in signature construction, so that differential expression analysis factors out similarities and focuses on the important mechanistic differences. In the analysis of breast cancers, the direct comparisons of Her2E and Luminal samples makes sense in this context as many of the genomic features are similar (eg expression of functional ERalpha, PR, and Her2). The major difference coming from up-regulation of Her2 vs ERalpha signaling. This same approach has been used by the TCGA consortium to characterize Her2E cancers. When we compare directly the two subtypes, the general molecular features of a breast cancer are factored out and the molecular features identified are directly related to the molecular drivers of the cancer in two subtypes which can, and do in clinical practice, serve as primary targets for pharmacological intervention.

The “standard” way of constructing a disease transcriptional signature that would compare tumor samples to the control breast tissue actually fails in this situation. In the revised manuscript we provide the analysis of this type of signatures (new Supplemental results 5). We show that signatures are dominated by the general common features of different breast cancer subtype are not very informative about the driver mechanisms in the two subtypes: high ER α activity in Luminal A cancers and high Her2 signaling in Her2-enriched cancers. These points are also discussed in the revised Results (second paragraph in the Use case 2 section).

Minor comments

- Page 3 of Supplemental Methods, in the last equation, the second d-bar should have a superscript “L”.

Answer: We corrected this error.

- Parts of Supplemental Results can be moved to the main text, which will make things look more concrete and easier to follow. I strongly urge the authors to redistribute the materials within the word limit. For example, they can move some of the lengthy enumeration of tools implemented in iLINCS from the main text to Supplemental Methods.

Answer: , Agreed. We have re-written the Results and Discussion section, and modified and added new figures. We hope that results are now exposed better and easier to follow.

- Page 4 of the main text, in the first line, “lead to”  “leading to”?

Answer: We corrected the error

- Page 4 of the main text, in the third last line, “network analyses approaches”  “network analysis approaches”?

Answer: We corrected the error.

- Have you considered incorporating CPTAC proteomics data into the backend? Of course, these data are in much smaller sample sizes due to the limited throughput of MS-based

proteomics, but it has novel components such as phosphoproteomics that RPPA does not offer with a sufficiently wide coverage of the proteome. There may be some value in including their data.

Answer: We completely agree and are actively working on a pipeline to systematically ingest CPTAC data. We discuss this along with multi-omic extensions as the major opportunity for improving iLINCS in the Discussion section (third and fourth paragraph).

- Once again, I would encourage the authors to think about the possibility of spurious association propagating through signature similarity-based analysis, or at least discuss this aspect in the Discussion.

Answer: This point is well taken. In the revised manuscript we, expanded the analysis and discussions in Use cases 1 and 2 to illustrate potential pitfalls and analysis workflow in iLINCS that can be used to address these issues. In addition to the pitfalls associated with overlapping pathways we discussed in Use case 1 (as elaborated above), we expanded the analyses in Use case 2 that is addressing issues which could arise in similarity-based analysis due to hierarchical nature of gene regulation. We also introduce the commentary about possible pitfalls in the discussion of the similarity-based analysis (Discussion section, fifth paragraph).

The hierarchical nature of gene expression regulation may lead to similar transcriptional signatures being generated by perturbing genes at different levels of the regulatory programs (eg signaling proteins vs transcriptional factor). For example, perturbations of different signaling proteins leads to modulation of the proliferation rates in cancer cell lines. Therefore, it may be expected that resulting transcriptional signatures share some similarities related to up- and down-regulation of, for example, cell cycle genes, although they may regulate very different sets of biological processes. At the same time, the signatures corresponding to perturbation of proteins regulating similar sets of sets of biological processes are likely to exhibit a higher level of similarity. The expanded analysis of top 100 chemical perturbagen signatures negatively correlated to Her2-enriched breast cancer signature in Use case 2, reveals that they all contain the “proliferation” component. However, the subset of most highly correlated signatures are also associated with mTOR inhibition indicating that the proliferation is modulated by modulating mTOR signaling. These fine points can be assessed by scrutinizing the gene expression profiles of genes in signature identified in CMAP analysis as shown in Figure 6. The associations with perturbations that modulate cellular proliferation, while real, could also be considered spurious as it is non-specific. The association with mTOR signaling is more specific and provides a higher-level mechanistic explanation for differences in proliferation rates. In our opinion this kind of analysis, greatly facilitated by iLINCS workflows, would always be important in interpreting results of CMAP analysis.

Reviewers' Comments:

Reviewer #1:

Remarks to the Author:

The authors have addressed my major concerns.

Reviewer #2:

Remarks to the Author:

The authors have addressed all of my comments.

Reviewer #3:

Remarks to the Author:

The authors brought significant improvement to the contents in this revised manuscript. The versatility of the tool and the benefit of the CMAP approach capitalizing on extensive data resources both are well presented in the use cases. I would still recommend users to exercise caution when interpreting the CP signature-based identification of potential therapeutic agents, since the process of ranking / prioritizing therapeutic targets depends on far more complex factors than the mRNA signatures induced by the compounds (e.g. the third use case where the analysis was limited to the ACE2 expressing cell line only). Other than that, the stories were substantially easier to follow than the initial submission and the clear discussion of potential limitations is also appreciable. I have no further comments and would support publication of this article.

Minor comments

Line 137: consider rewriting "via CMAP analysis with the validation via analysis of EHR". Something is off in this sentence.

Line 280: "indicating..." – this sentence is a speculation. I would rather say "generating a hypothesis that the proliferative property of Her2E tumor can be countered by the inhibitors of Top21 and AURKA such as XXX".

Line 325: "TNA-alpha" \diamond TNF-alpha?

The details in the Methods section are largely disconnected with the results (understandably so). I do not have a good suggestion how to integrate them with the main results. Perhaps a large chunk of them could be provided in a separate supplementary information.

Response to reviewer's additional comments

Two out three reviewers had no additional comments. Here we address the Minor comments by Reviewer #3.

Reviewer #3 (Expertise: Computational and systems biology, OMICS integration):

Summary

The authors brought significant improvement to the contents in this revised manuscript. The versatility of the tool and the benefit of the CMAP approach capitalizing on extensive data resources both are well presented in the use cases. I would still recommend users to exercise caution when interpreting the CP signature-based identification of potential therapeutic agents, since the process of ranking / prioritizing therapeutic targets depends on far more complex factors than the mRNA signatures induced by the compounds (e.g. the third use case where the analysis was limited to the ACE2 expressing cell line only). Other than that, the stories were substantially easier to follow than the initial submission and the clear discussion of potential limitations is also appreciable. I have no further comments and would support publication of this article.

Minor comments

Line 137: consider rewriting “via CMAP analysis with the validation via analysis of EHR”. Something is off in this sentence.

Answer: We modified this sentence

Line 280: “indicating...” – this sentence is a speculation. I would rather say “generating a hypothesis that the proliferative property of Her2E tumor can be countered by the inhibitors of Top21 and AURKA such as XXX”.

Answer: We modified this statement which now reads: “...**suggesting** that such treatments **may** counter-act the Her2E tumor driving events...”, instead of the original statement which we agree was too speculative: “...**indicating** that such treatments **can** counter-act the Her2E tumor driving events...”

Line 325: “TNA-alpha” □ TNF-alpha?

Answer: We modified this particular section and this typo is no longer there.

The details in the Methods section are largely disconnected with the results (understandably so). I do not have a good suggestion how to integrate them with the main results. Perhaps a large chunk of them could be provided in a separate supplementary information.

Answer: We have already placed a large portion of methodological details in the supplemental document and think that current level of methodological detail in the main

text is needed to easily understand high-level structure of iLINCS and the analysis methods used in the use cases.